# MRI Contrast Agents in Glycobiology

**DOI:** 10.3390/molecules27238297

**Published:** 2022-11-28

**Authors:** Carlos F. G. C. Geraldes, Joop A. Peters

**Affiliations:** 1Department of Life Sciences and Coimbra Chemistry Centre, Faculty of Science and Technology, University of Coimbra, 3000-393 Coimbra, Portugal; 2CIBIT-Coimbra Institute for Biomedical Imaging and Translational Research, 3000-548 Coimbra, Portugal; 3Department of Biotechnology, Delft University of Technology, van der Maasweg 9, 2629 HZ Delft, The Netherlands

**Keywords:** magnetic resonance contrast agents, diagnostic agents, theranostics, biomarkers, lectins, selectins, asialoglycoprotein receptor, sialic acid, glycohemoglobin

## Abstract

Molecular recognition involving glycoprotein-mediated interactions is ubiquitous in both normal and pathological natural processes. Therefore, visualization of these interactions and the extent of expression of the sugars is a challenge in medical diagnosis, monitoring of therapy, and drug design. Here, we review the literature on the development and validation of probes for magnetic resonance imaging using carbohydrates either as targeting vectors or as a target. Lectins are important targeting vectors for carbohydrate end groups, whereas selectins, the asialoglycoprotein receptor, sialic acid end groups, hyaluronic acid, and glycated serum and hemoglobin are interesting carbohydrate targets.

## 1. Introduction

All cells in nature are covered with a dense array of glycans [1]. Because the constituting monosaccharides can be connected in many ways, the glycans are encoded with a lot of information. Their biosynthesis occurs post-translationally by a wide array of enzymes and their compositions respond highly dynamically to intrinsic and extrinsic signals by undergoing rapid enzyme-mediated changes. Many diseases are associated with alterations in the expression of particular carbohydrates. Glycoprotein-mediated interactions are at the heart of medically relevant processes, including immune response, viral infection, inflammation, bacterial adhesion, metastasis, and reproduction. The main proteins involved are lectins, antibodies, and enzymes. Visualization of these interactions and the extent of expression of the sugars are thus useful tools in medical diagnosis, monitoring of therapy, and drug design.

During the last decades, the progress in science and technology led to the introduction of many new noninvasive bioimaging modalities, including X-ray computed tomography (CT), magnetic resonance imaging (MRI), optical imaging (OI), positron emission tomography (PET), single-photon emission computed tomography (SPECT), and ultrasound [2]. Most of these techniques have also been exploited for the study of sugar-protein interactions. Contrast agents (CAs) have been developed to target sugars that are biomarkers of disease with probes having either natural or synthetic targeting vectors and vice versa CAs to which sugars are attached to target protein receptors involved in the relevant sugar-protein recognitions. The present review focuses on the application of Gd^3+^-chelates and paramagnetic and superparamagnetic nanoparticles (NPs) as CAs in studies related to interactions between carbohydrates and selectins. In addition, the exploitation of these CAs as diagnostic tools targeted at carbohydrates, such as glycated serum and hemoglobin are included.

Various polysaccharides or fragments thereof, including hyaluronic acid and chitosan, are widely used as platforms for paramagnetic CAs in diagnosis and image-guided drug delivery. These topics are covered in several excellent recent reviews [3,4,5,6] and therefore will not be discussed in the present review. MRI using glucose-based chemical exchange saturation transfer (glucoCEST/CESL) is developing as an interesting highly sensitive and metal-free diagnostic imaging procedure. For an overview of this area, we refer to another excellent recent review [7].

## 2. Relaxivity of MRI Contrast Agents

MRI is based on nuclear magnetic resonance (NMR) principles. The contrast in the images is generated by differences in intensities of NMR resonances, usually of water protons. Local ^1^H concentrations, longitudinal (*R*_1_ = 1/*T*_1_) and transverse (*R*_2_ = 1/*T*_2_) relaxation rates, and diffusion govern these intensities and, consequently, the contrast can be enhanced by the application of CAs modulating these parameters [8,9,10,11,12,13,14]. Today, the most frequently applied CAs are paramagnetic compounds that improve the contrast by enhancing relaxation rates. The efficacy of CAs is usually expressed by the relaxivity, the longitudinal, or transverse relaxation rate enhancement normalized for a solution containing 1 mM of paramagnetic metal ions (*r*_1_ and *r*_2_, respectively). Those with a ratio *r*_2_/*r*_1_ smaller than about 10 can produce bright spots in *T*_1_-weighted (*T*_1w_) images and are called positive or *T*_1_ CAs. The presently most popular clinically applied CAs are Gd^3+^-complexes of low molecular weight octadentate ligands with one or two water molecule(s) in the first coordination sphere of Gd^3+^ (Figure 1) [15,16].

The *r*_1_ of such metal chelates as a function of the external magnetic field strength (*B*_0_), the ^1^H nuclear magnetic relaxation dispersion (NMRD) profile, can be evaluated with models based on the Solomon-Bloembergen-Morgan and the Freed theories for the contributions of the water protons in the inner-and outer-sphere of the metal ion, respectively [2,8,9]. The main parameters governing *r*_1_ are the number of water molecules in the first coordination sphere of the metal ion (*q*), their residence time in the first coordination sphere (*τ*_M_ = *k*_ex_*^−^*^1^, where *k*_ex_ is the water exchange rate), and the molecular tumbling time (*τ*_R_) (Figure 2). Sufficient thermodynamic and kinetic stability is usually only possible for *q* = 1, although a few potential CAs with sufficient stability have *q* = 2 or 3. Usually, *r*_1_ has an optimum around *τ*_M_ = 10 ns and *r*_1_ rises with increasing *τ*_R_ up to about *τ*_R_ = 10*^−^*^8^ s, after which *r*_1_ becomes almost independent of *τ*_R_.

*T*_2_ or negative CAs increase the *R*_2_ of water protons significantly more than the *R*_1_ and give rise to dark spots in *T*_2w_ or *T**_2w_ images. The main representatives of this type of CAs are the superparamagnetic iron oxide nanoparticles (NPs). The relaxivity of these NPs has also inner- and outer-sphere contributions. The inner-sphere contributions arise from protons that exchange between the surface groups and the bulk and generally can be modeled by the Solomon-Bloembergen-Morgan equations. The inner-sphere relaxivity is often negligibly small because only the paramagnetic ions at the surface of the NPs, which are in direct contact with water, play a significant role. The outer-sphere relaxivity, which is due to water protons diffusing along the particle, is usually much larger and is governed by the volume fraction of the superparamagnetic particles (*υ*), the diffusion correlation time (*τ*_D_) as defined in Equation (1), where *d* is the diameter of the particle and *D* is the diffusion coefficient), and *M* the magnetization of the NP. The latter reaches usually its saturation value *M*_S_ at *B*_0_ ≤ 0.5 T, which is the lower limit for common clinical equipment. Hence, *R*_2_ of superparamagnetic NPs generally is independent of *B*_0_. Several models have been developed to describe the transverse relaxivity of spherical superparamagnetic NPs [17,18,19,20,21]. Three regimes can be distinguished with limits defined by *τ*_D_, the static correlation time (Δ*ω*^−1^) as defined in Equation (2), and the correlation time *τ*_CP_ (half the time interval between successive 180° pulses in a Carr-Purcell-Meiboom-Gill pulse sequence). For small particle sizes (*τ*_D_ << Δ*ω*^−1^), the condition for the motional averaging regime (MAR) is satisfied, which implies that diffusive motions are faster than the spatial variation of the local field inhomogeneities produced by the individual particles resulting in a motional averaged effect by these particles. Then, *R*_2_ and *R*_2_* are linearly dependent on *τ*_D,_ *M*_S_^2^_,_ and *υ*. When *τ*_D_ > 2.72Δ*ω*^−1^, *R*_2_ and *R*_2_* reach their maximum value and the static dephasing regime (SDR) is entered, where, *R*_2_ and *R*_2_* are solely dependent on *M*_S_ and *υ*. For larger NPs, *R*_2_* as a function of *τ*_D_ remains constant, whereas *R*_2_ ultimately decreases with *τ*_CP_ because then the magnetic field gradients close to the NPs become so strong that refocusing the magnetization is impossible.
(1)$$\tau_{D}=\frac{d^{2}}{4D}$$
(2)$$\frac{1}{\Delta \omega }=\frac{3}{\gamma \mu_{0}M_{S}}$$

Iron oxide particles can be divided into three categories: (i) ultra-small iron oxide superparamagnetic NPs (USPIOs) with hydrodynamic sizes less than 50 nm, (ii) superparamagnetic iron oxide NPs (SPIOs) with hydrodynamic sizes larger than 50 nm, and (iii) micron-sized NPs (MPIOs). These NPs are usually used as negative CA. However, very small USPIOs with hydrodynamic sizes less than 5 nm may exhibit sufficiently small *r*_2_/*r*_1_ ratios at 0.5–1.5 T to be suitable for application both as *T*_1_ or dual *T*_1_–*T*_2_ CA [11].

Another MRI approach exploits ^1^H chemical exchange saturation transfer (CEST) of mobile protons of biomolecules, such as -OH and -NH [22]. Low-power radio frequency pulses at the resonance frequency of labile protons transfer magnetization to the bulk water protons leading to attenuation of the NMR signal. This phenomenon improves the detectability of labile protons and can be switched on and off at will through the saturation pulses. The CEST sensitivity is optimal with high exchange rates, as long as the resonances for exchangeable protons and bulk water do not coalesce, which implies that the difference in resonance frequencies of the labile substrate protons and water (Δ*ω*) must be larger than the exchange rate (Δ*ω* ≥ *k*_ex_). Paramagnetic Ln^3+^-complexes (Ln ≠ Gd) can be applied to increase Δ*ω* [23,24]. The magnitude of Δ*ω* is strongly dependent on the choice of the Ln^3+^ ion, those with the highest effective magnetic moments (Dy^3+^, Ho^3+^, Tm^3+^) induce the largest shifts. When the exchangeable protons are incorporated into the Ln complex, the method is called PARACEST.

MRI has a significantly higher spatial resolution (μm) than radiodiagnostic imaging techniques (mm), but its use for in vivo imaging of biomarkers is hampered by the low intrinsic sensitivity. For the currently applied low molecular weight Gd-complexes, a relatively high local concentration of CA is required (about 10^−5^ M) to achieve sufficient contrast enhancement in *T*_1w_ images. Radionuclide imaging modalities such as PET (Positron Emission Tomography), SPECT (Single Photon Emission Computerized Tomography) (10^−11^–10^−12^ M), and Optical Imaging (OI) (10^−15^–10^−17^ M) are much more adequate in this respect. The amount of a particular receptor is typically in the order of magnitude of 10^5^ per cell, which corresponds to a local CA concentration in the nM–μM range on a cell volume basis [25,26,27,28]. A strategy to overcome the problems related to the low intrinsic sensitivity of MRI is to apply vectorized CAs, which deliver a high payload of a paramagnetic compound to the site of interest. For lanthanide-based CAs, this has been achieved mostly by using nanosized materials loaded with paramagnetic ions, including micelles, liposomes, perfluorocarbon NPs, nanogels, and solid NPs. Obviously, it is desirable that the main parameters governing the relaxivity of the individual paramagnetic ions (*q*, *τ*_M_, *τ*_R_) are optimized [29,30]. An important way of amplification is receptor-mediated endocytosis. The efficacy depends on the localization of the CAs entrapped in the cell. For example, Gd-HPDO3A internalized into rat hepatocarcinoma cells exhibits higher *R*_1_ values in the cytoplasm than in intracellular vesicles [31], where the relaxivity is limited by the water exchange rate across the sub-cellular membranes. NPs can be selectively entrapped in tumors passively as a result of the leaky neovasculature, which allows NPs to pass the endothelium using the enhanced permeability and retention (EPR) effect [32].

Most *T*_2_ CAs applied in this field are superparamagnetic NPs for which each of the metal ions has a very high *r*_2_ (typically 60–400 s^−1^ mM^−1^). Usually, it concerns superparamagnetic NPs that have reached the saturation magnetization at the *B*_0_ values of the current clinical MRI equipment, and then *r*_2_ is independent of *B*_0_. The interpretation may be hampered by difficulties in distinguishing the CA-induced darkening from partial-volume artifacts, motion artifacts, and tissue inhomogeneities.

For PARACEST CAs multiplication of the effect can be achieved by, for example, attachment to dendrimers, and particularly by inclusion in liposomes or cells. The high number of water molecules entrapped inside these vesicles provides high sensitivity, lowering the detection threshold to the pico/femtomolar range in terms of vesicle concentration [33,34].

## 3. Glyconanoparticles

The recent advances in the tailored synthesis of glycosylated NPs (glyconanoparticles or GNPs), due to a combination of nanotechnology with glycobiology, allowed the enormous growth of a variety of biomedical applications in which they work as mimetics of natural glycoconjugates [35,36,37]. The control of the shape, size, and organization of the multivalent sugar shell around stable NPs led to the production of structured glycosylated SPIOs, metal (Au, Ag, Cu) NPs, glycosylated quantum dots (QDs), fullerenes, single-wall carbon nanotubes (SWCNTs), and self-assembled GNPs using amphiphilic glycopolymers or glycodendrimers [38], some of which have been proposed as MRI contrast agents.

A variety of oligosaccharides has been used as coatings for the stabilization of SPIOs by preventing their agglomeration and promoting their solubility in aqueous and biological media. Among the many polymeric coating materials proposed are cellulose, chitosan, pullulan, dextran and derivatives including carboxymethylated dextran, starch, arabinogalactan, and glycosaminoglycan (GAG) [39,40]. Chemical modification of dextran affects the formation and stability of SPIOs and USPIOs. Reduction of the terminal reducing-sugar influences the particle size, coating stability, and magnetic properties (and thus their relaxivity) [41]. USPIOs with different applications were obtained by iron core surface binding of aminopropyl silane groups (silanization), followed by covalent conjugation with partially oxidized dextran and reduction of the Schiff base. The in vivo biodistribution of the NPs in general is determined by coating modifications, e.g., by hiding their electrical surface charge [42], or by PEGylation that protects the NPs from recognition and endocytosis by the liver reticuloendothelial system (RES), for example in Clariscan^®^ [43].

Many targeted SPIOs have been reported with coatings conjugated with appropriate targeting vectors, some of which are described in other sections of this review (see, e.g., Section 6. on asialoglycoprotein receptor targeted CAs). Here, several examples of tumor theranostic applications of sugar-coated iron oxide are described. USPIOs coated with mannose, ribose, and rhamnose proved to be an efficient targeting system for theranostic applications, with very good *r*_2_ values and large heat release upon application of frequency (RF) radiation with amplitude and frequency close to the human tolerance radio-limit, making them promising as negative MRI contrast agents and for magnetic fluid hyperthermia (MFH) [44]. The carbohydrate coating provides targeting properties to the GNPs. In particular, rhamnose showed a high affinity to skin lectin [44]. An example of tumor-targeting SPIOs is given by SPIOs coated with amphiphilic compounds based on 1-oleyl 2-acetamido-2-deoxy-d-glucopyranoside. Aqueous suspensions of these materials are very stable owing to a negative zeta potential and show favorable characteristics for applications as *T*_2w_ MRI CAs (*r*_1_ < 4.5 s^−1^ mM^−1^ and *r*_2_ 140–200 s^−1^ mM^−1^ at 1.5 T, 37 °C). Furthermore, these NPs, with a hydrodynamic diameter of about 50 nm, have antimitotic activity as studied on rat glioma (C6) and human lung carcinoma (A549) cell lines, showing equal or even better anti-tumor effects relative to the free glycosides [45].

A platform for image-guided efficient doxorubicin (DOX) delivery was constructed by embedding SPIO clusters (10–15 nm size) in a hydrogel-like crosslinked plant galactoxyloglucan PST001. Folic acid (FA) groups were conjugated to surface OH-functions. The glucan has galactose end groups that have affinity to tumor selectins. (FAP-IONPs). The NPs had a hydrodynamic size of about 289 nm and a zeta potential of −11 mV. An aqueous solution of the NPs was stable for 120 days and the NPs showed less than 10% DOX release at pH 7.4, but at the slightly acidic values (pH 5.5–6.5) that often occur in the vicinity of tumors, the release rate enhanced steeply. Since the pH in the vicinity of tumors often is slightly acidic, the release in cancerous tissue is preferred. The tumor selectivity and successful DOX delivery were demonstrated on human cervical carcinoma (HeLa) tumor-bearing xenograft nude mice. *T*_2w_ MRI (1.5 T) images showed large negative contrast. [46].

A theranostic agent for combined *T*_1w_ magnetic resonance angiography and anti-tumor therapy by anti-heparanase has been developed by coating very small USPIOs with heparin and fractionated heparins. Optimal results were obtained with a depolymerized heparin fraction (MW_avg_ < 8 kDa), which gave well-dispersed iron oxide cores, with diameters of around 4.6–5.4 nm (TEM), hydrodynamic sizes in water (DLS) of 29.4–55.7 nm and negative zeta potentials (−40 to −51 mV) at physiological pH, ensuring their excellent colloidal stability. These NPs have a ratio *r*_2_/*r*_1_ (*r*_1_ = 4.0; *r*_2_ = 12.8 s^−1^ mM^−1^ at 37 °C and 1.5 T) that makes them suitable for application as positive CA. These NPs generated images with excellent anatomical details depicting carotids, aorta, heart chambers, main veins, and even some smaller vessels. The bright signal in the vascular system was maintained for more than 210 min post-injection. Together with their heparanase inhibition for antitumoral treatment, these NPs have potential as theranostics [47].

Besides tumor theranostics, carbohydrate-coated iron oxide NPs have shown other important applications such as MRI CAs. For example, novel mannose-labeled SPIOs were developed to detect sentinel lymph nodes (LN) by MRI. These NPs with a hydrodynamic size of 73.9 nm, consist of maghemite iron oxide (γ-Fe_2_O_3_) SPIOs sterically stabilized by two block copolymers: a stabilizing polymer (70%) and a macrophage-targeting mannose-polymer (30%). They were successfully tested in pre-operative MRI imaging (3.0 T) and intraoperative magnetometer detection was carried out using a large animal model (anesthetized white pigs). The results show the potential of the technique to overcome the limitations of using sentinel lymph node biopsy in cancers of the head and neck, which, despite their clinical and diagnostic value, is adopted only in limited cases due to concerns about the detrimental consequence to survival of false negative results in a highly curable setting [48]. In another example, d-mannose-coated maghemite NPs (d-mannose(γ-Fe_2_O_3_) were demonstrated to label neural stem cells (NSCs) much better than the uncoated NPs, the labeled cells were visualized by ex vivo MRI and their localizations were confirmed by histological sections. The progenitor properties and differentiation of the NSCs were not affected by the labeling, although changes in cell proliferation, viability, and neurosphere diameter were observed at higher NPs concentrations. d-mannose coating of the NPs improved NSCs labeling and was efficiently detected by ex vivo MRI *T*_2_ maps (9.4 T) of mouse brain slices, confirmed by Prussian blue staining and immunohistochemistry [49].

Sugar-coated Au GNPs have also been used as water-soluble, biocompatible, and non-cytotoxic nanoplatforms for targeted MRI CAs. Many hybrid GNPs having the same gold nanoplatform sugar conjugates and Gd^3+^ chelates have been studied [50], for instance, paramagnetic gold GNPs with different ratios of thiol-ending sugar (glucose, galactose, or lactose) conjugates and Gd^3+^ thiolated N-alkyl DO3A (DO3A = tetraazacyclododecane triacetic acid (DO3A) ligands. Glycoconjugates of d-glucose (glcC_2_S and glcC_5_S), d-galactose (galC_5_S), and β-lactose (lacC_5_S) having a –(CH_2_)_n_–S– (*n* = 2,5) linker were shown to have higher *r*_1_ values than Gd-DTPA (Figure 3a). In vivo MRI of glioma (generated with GL261 tumoral cells) in mice indicated that, at the same Gd^3+^ concentration, glcC_5_S-GNPs enhance the positive contrast in the tumoral zones better than CAs in clinical use (Figure 3b) [51]. A simple, cost-effective high-throughput method for selecting such hybrid AuGd-GNPs for application as CAs in in vivo studies was developed [52]. This method used their post-mortem ex vivo relative contrast enhancement, which did not correlate well with their respective in vitro relaxivities. The results obtained with different AuGd-GNPs suggested that the linker length of the sugar conjugate could modulate the interactions with cellular receptors and therefore the relaxivity value. However, the ex vivo method could produce an underestimation of the actual contrast enhancement potential of the AuGd-GNPs, as shown by in vivo animal MRI studies.

Recently, novel hybrid gold nanoparticles, decorated with Gd-DOTA and stabilized by electrostatic adsorption of a lactose-modified chitosan polymer (CTL; Chitlac) (Gd-DOTA-IN-CTL-Au NPs), were developed and investigated as an MRI-based theranostic nanoplatform. These NPs are spherical (18 nm diameter by TEM) and stable at physiological pH, as shown by zeta potential and DLS measurements, nontoxic towards Mia PaCa-2, TIB-75, and KKU-M213 cell lines, and very efficient as photothermal therapy (PTT) agents towards those cancer cells. Preliminary in vivo MRI (3 T) studies, using male tumor-bearing BALB/cA nude mice injected at the tail vein with the NPs, showed a clear bright contrast of the tumor region 30 min after injection, on *T*_1w_ images, indicating the potential of the NPs as a theranostic agent, combining and PTT therapeutic properties in the same nanoplatform [53].

## 4. Plant Lectins as Probes

Lectins are proteins that are broadly found in animals, plants, and lower organisms. They specifically bind oligosaccharide moieties of glycoproteins and glycolipids on cell membranes. The specificities and the affinities are determined by various factors including monosaccharide composition, shape, and density of the glycans. In molecular imaging, lectin-carbohydrate interactions can be exploited by applying either lectins or carbohydrates as targeting vectors.

Several plant lectins (agglutinins) discriminate between cell types, most likely through lectin-carbohydrate interactions. They are widely available and therefore are attractive targeting groups for MRI CAs. For example, they have been exploited to monitor the migration of transplanted rat fetal brain tissue by labeling them with superparamagnetic ferrite particles (diameter 1 μm) covalently coupled to wheat germ agglutinin (WGA) [54], which is known to bind to *N*-acetyl-d-glucosamine and sialic acid (SA). These particles appeared to form aggregates on the outside of the cells.

Tomato lectin, LEA, is poly-specific: it interacts with several structurally different glycans; not only with plant glycoproteins but also with animal and human glycoproteins, particularly with erythrocytes and endothelial cell surfaces [55]. A series of latex NPs (diameters 100–900 nm) has been derivatized with LEA and subsequently, lysines of LEA were coupled through amino groups to a carboxylate of Gd-DTPA. The resulting NPs were tested as *T*_1_ CA to visualize blood vessels in murine liver and human placental cotyledon [56,57]. LEA has also been attached to mixed oligomers of Gd-DTPA and diols or diamines, and as LEA-DTPA-Gd conjugate to chitosan and bovine serum albumin (BSA) NPs. The resulting materials have been proposed as alternatives for blood pool CAs because they have a high affinity to the inner surface of blood vessels as well [57,58,59,60]. A theranostic agent prepared by conjugation of LEA to glycerol mono-oleate coated SPIOs has been exploited to deliver adsorbed cytostatic paclitaxel into K562 leukemia cells [61]. These NPs were applied simultaneously as *T*_2_ CAs.

Dual-mode CT/MRI tumor-targeting CAs have been constructed by covalent attachment of the lectins WGA, Ricinus Communis Agglutinin (RCA_120_), and concanavalin A to core–shell superparamagnetic Fe_3_O_4_@Au NPs (diameter 22 nm) [62]. The lectins were attached through a bivalent NHS-PEG-S-S-PEG-NHS linker. The WGA-Fe_3_O_4_@Au NPs had an *M*_S_ value of 8.7 emu g^−1^ at 25 °C, which is about 28 emu g^−1^ lower than that of the bare Fe_3_O_4_ core due to the covering diamagnetic shells. The *r*_2_ of the CA was 56.12 s^−1^ mM^−1^ at 1.5 T and 25 °C. These lectin-Fe_3_O_4_@Au NPs produced satisfying images of colorectal cancer in BALB/C nude mice by CT and *T*_2w_ MRI.

## 5. Targeting of Selectins

Selectins are a special type of lectins that constitute a family of cell adhesion trans-membrane proteins with a key role in the cascade of events taking place at endothelial cells upon activation by the immune system [63]. They mediate the cell trafficking of leukocytes to an injured or diseased site by rolling along blood vessel walls through protein-carbohydrate interactions. The family can be subdivided into E-, P-, and L-selectins, which share primary and secondary structural homology in their Ca^2+^-dependent *N*-terminal lectin moieties that contain the carbohydrate recognition domain (CRDs). The main ligand recognized by these CRDs is the tetrasaccharide sLe^x^, which binds them non-covalently and reversibly. L-selectins are constitutively expressed on leukocytes, whereas E-selectins are recruited on endothelial cells of the lumen of blood vessels upon stimulation of the immune system. P-selectins are expressed on activated platelets and endothelial cells. They bind to sLe^x^ and sulfated tyrosine residues on the *N*-terminal region of the P-selectin glycoprotein ligand-1 on the surface of leukocytes. (Figure 4). The density of selectins is upregulated substantially induced by injury or disease. Selectins are suitable as markers of inflammation associated with diseases including cardiovascular disorders, cancer, and rheumatoid arthritis. Interestingly, the upregulation of endothelial E-selectins can be detected on the blood side of the blood–brain barrier (BBB) in response to a lesion on the brain side, which is not easily accessible for CAs when the BBB is intact.

Targeting vectors for CAs suggested for the molecular imaging of upregulated selectins include antibodies or their immunospecific fragments (Fab), aptamers, peptides, or small-molecule peptidomimetics emerging from phage display or small-molecule screens [64]. Here, we will focus on glycoconjugate probes and targets. Based on a model of the interaction between E-selectin and sLe^x^, a synthetically easily accessible sLe^x^-mimetic (sLe^x^m, Figure 5) was designed that has potency as an inhibitor of the parent tetrasaccharide [65]. Therefore, Muller c.s. have bound sLe^x^m via a propylamine linker to Gd-DTPA to give bisamide Gd-DTPA-(sLe^x^m′)_2_ [66]. Physico-chemical characterization by ^1^H and ^17^O NMR showed that Gd-DTPA-(sLe^x^m′)_2_ has *q* = 1 and about the same *r*_1_ as that of Gd-DTPA [67]. Extensive animal studies demonstrated that this CA targets selectively vascular endothelial E-selectin in the brain (with intact BBB), liver, or spleen after inducing inflammation, which results in modest but visible contrast enhancement of the affected vasculature [68,69,70] The sLe^x^-mimetic has also been coupled to the dextran coating of USPIOs with a core diameter of 5–6 nm (USPIO-sLe^x^m’) [71]. In vivo and in vitro MRI investigations both showed that USPIO-sLe^x^m′ interacts with endothelial E-selectin. At Larmor frequencies above 1 MHz, these systems have *r*_2_ >> *r*_1_ (*r*_2_ = 78.6 s^−1^ mM^−1^ at 60 MHz, 37 °C). Therefore, they are particularly suitable as negative CAs in *T*_2w_ or *T*_2w_* images. An additional advantage of the coupling to USPIOs is that each sLe^x^-mimetic function can deliver a high payload of paramagnetic ions per USPIO particle to an E-selectin ligand leading to an amplification of the relaxation enhancement. Furthermore, these materials allow multivalent and thus stronger interactions with the target selectin. The same mimetic (sLe^x^m′) has been attached to carboxylated USPIOs that were also coated with PEG_750_ to prolong the plasma circulation time and minimize the nonspecific accumulation of iron oxide in tissues [72]. Comparison with similar measurements using particles without the sLe^x^-mimetic targeting vector showed that the darkening in the image after administration of the targeted particles was twice as large. A similar system but now with PEG_2000_ as linker between the USPIOs and sLe^x^ (USIO-PEG_2000_-sLe^x^) had a particle diameter of 53 nm with *r*_1_ = 9.8 s^−1^ mM^−1^ and *r*_2_ = 29 s^−1^ mM^−1^ at 1.4 T and 37 °C [73]. In vivo experiments demonstrated the detection of E-selectin in vivo in nude mice that had undergone nasopharyngeal carcinoma (NPC) metastasis.

Van Kasteren et al. have synthesized a series of dextran-coated iron oxide NPs decorated with oligosaccharides of increasing carbohydrate complexity [74]. Only NPs conjugated with sLe^x^ itself, USPIO-sLe^x^ (Figure 5) targeted activated brain endothelium. With animal models, it was shown that these particles are sensitive negative CAs for early detection of endothelial activation by brain disease events. Such lesions were not detectable with conventional MRI.

A dual probe (MRI/fluorescence) mimicking the P-selectin binding of PSGL-1 (carrying sLe^x^ and sulfated tyrosines) has been prepared from carboxymethylated polydextran (*M*_w_ 27,000, degree of substitution, DS = 0.84) [75]. After attachment of sulfate groups (DS = 0.41), Gd-DOTA units (DS = 0.08), and a fluorescein isothiocyanate label (DS = 0.008) a material CM8FS (see Figure 6) was obtained that in cytometry experiments on whole human blood cells and on platelets specifically interacted with activated platelets. No binding to other blood cells or resting platelets was observed. Activated platelets incubated with CM8FS produced a bright *T*_1w_ MR image and were successfully tested in vivo for the MRI location of inflammatory vascular tree lesions in ApoE^−/−^ mice [75,76]. The material lacking the sulfate groups gave a much weaker vascular contrast enhancement.

Just SA can also be used to target selectins. This has been demonstrated by Fan et al., who covered SPIOs with mesoporous polydopamine (MPDA@SPIO) bound via Fe-catechol coordination followed by loading with a mixture of a conjugate of polyethyleneimine and sialic acid (SA-PEI) and α-fetoprotein regulated ferritin gene (AFP-Fth) [77]. The various components of MPDA@SPIO/SA-PEI/AFP-Fth were kept together by electrostatic interactions. The SA targets E-selectin on hepatocellular carcinoma and promoted endocytosis, whereas transfection of AFP-Fth induced a significant upregulation of the expression level of ferritin. In this way, the endogenous contrast was enhanced in *T*_2w_ images. After chelation of another dopamine-based system, SA-PEG-MPDA@SPIO with Fe^3+^ and loading it with DOX, a nanoplatform SA-MPDA@SPIO/DOX/Fe^3+^ with good *T*_1_ and *T*_2_ enhancing properties (*r*_1_ = 4.3 s^−1^ mM^−1^; *r*_2_ = 106 s^−1^ mM^−1^ at 3 T and 37 °C) was obtained that may be used for image-guided combined chemo and photothermal therapy [78].

SAs are abundant in brains, for example in ganglioside GM1, which has been found bound to β-amyloid peptide (Aβ) in the brains of Alzheimer’s disease patients [79]. This inspired Kouyoumdjian et al. to use SA attached through a linker to dextran-coated iron oxide NPs (USPIO-SA, core diameter about 5 nm, see Figure 5) for the ex vivo detection of Aβ [80]. NPs coated with SA attached to BSA were able to pass the BBB [81]. The affinity of these NPs for Aβ plaques was demonstrated with several techniques and could be visualized with *T*_2w_* MRI of a mouse brain.

Since the contrast enhancements achieved by the above-mentioned CAs are generally modest, micro-sized iron oxide particles (MPIOs) with diameters in the range 0.9–8.5 μm, have been proposed as alternative platforms for targeted vectors [82,83]. These particles can deliver much more magnetic material per particle. Since for such large sizes, it may be expected that *τ*_D_ > 2.72Δ*ω*^−1^, *R*_2_ will be in the SDR, which implies that *r*_2_* is maximal and independent of the particle size [84,85]. It should be noted that the particles are so big that *r*_2_ is in the partial refocusing regime, where *r*_2_ is dependent on *τ*_CP_ [86]. Practically, this generally means that *r*_2_ << *r*_2_*, and thus these particles are most suitable for gradient echo-weighted images. In addition to having high *r*_2_* values, the MPIOs can deliver payloads of iron per particle orders of magnitude larger than for USPIOs. Moreover, thanks to the so-called blooming effect, the hypo-intensive spot in an MRI image of an MPIO extends to a distance of at least 50 times its physical diameter [85]. Due to their large size, MPIOs usually remain intravascular and have a very short blood half-life (45–100 s). Therefore, they produce images with a high target-to-background ratio of endovascular targets immediately after administration [64,87]. By contrast, USPIOs have blood residence times of 24–48 h leading to unfavorable target-to-background ratios, particularly immediately after injection. MPIOs with attached antibodies have, for example, been used for the imaging of endothelial cell adhesion molecules VCAM-1 and P-selectin in mouse models of atherosclerosis and of cerebral ischemia [88,89,90].

Sulfated polysaccharides including sulfated dextran, heparin, and fucoidan are capable of interacting with P-selectin. Interaction with the brown seaweed-based sulfated polysaccharide fucoidan (a polysaccharide with a backbone of (1→3 and 1→4)-linked α-l-fucose having sulfate groups at O-2 and O-3) appeared to be the most efficient [91,92]. Therefore, targeting P-selectin by fucoidan-based nanoparticles and microparticles has been used as a new approach for imaging various inflammatory processes in which P-selectin is involved, including atherothrombosis and abdominal aortic aneurysms (AAAs). The adhesion of the inflamed leukocytes to the aorta endothelium has been mimicked with MPIOs functionalized with low-molecular-weight fucoidan (LMWF). In a mouse model, the expression of P-selectin in AAA was imaged with this *T*_2_ CA [93]. Dextran-coated USPIOs with covalently attached to low molecular weight fucoidan have provided an MRI CA with both high *r*_1_ and *r*_2_ (15.2 and 137.4, respectively, at 1.42 T and 37.4 °C). In vivo experiments demonstrated that the resulting agent localizes specifically in the thrombus area and has an affinity to P-selectin [94].

A *T*_1_ nanoparticulate CA (diameter 244 nm) constructed by self-assembly of the cell-penetrating cationic low-molecular-weight protamine (TPP_1880_), the anionic low-molecular-weight fucoidan (LMWF_8775_), and the anionic CA Gd-DTPA has been applied for in vitro imaging of P-selectins in HUMAC cells [95]. Recently, P selectin-targeted submicron particles (diameter 0.7–0.8 μm) were constructed by an emulsion crosslinking of fucoidan with dextran that was grafted with Gd-DOTA. The resulting compound had *r*_1_ = 6.7 s^−1^ mM^−1^ and *r*_2_ = 37.6 s^−1^ mM^−1^ at 7 T [96].

## 6. Targeting the Asialoglycoprotein Receptor

The hepatic asialoglycoprotein receptor (ASGP-R) is an organ-specific hetero-oligomeric C-type lectin, consisting of a major 48 kDa (ASGP-R1) and a minor 40 kDa subunit (ASGP-R2). This transmembrane protein is expressed primarily at the sinusoidal surface of the liver hepatocyte cells, which recognizes terminal β-galactosyl and β-*N*-acetyl-galactosaminyl residues on de-sialylated glycoproteins [97]. It plays a critical role in serum glycoprotein homeostasis, as when these residues are exposed upon desialylation, they are recognized and bound by ASGPR. Then, receptor-mediated endocytosis and lysosomal degradation removes the concerned glycoproteins. [98]. The capacity of the hepatic ASGP-R to recognize these residues on de-sialylated glycoproteins can be used for liver-specific drug delivery and targeting of artificial glycoconjugates [99]. This has been successfully achieved with galactose/lactose-containing glycoconjugates, with a multivalence effect (tetra > tri > di > mono) on their liver uptake [100]. The ASGP-R may also facilitate hepatic infection by multiple viruses including hepatitis B [101].

The functional imaging of liver ASGP-R has both diagnostic and prognostic values during the treatment of liver pathologies (e.g., cancer, hepatitis B). A few hepatocyte-specific MRI CAs are currently available for the detection of hepatic lesions, such as the clinically accepted Gd^3+^ complexes of lipophilic or amphiphilic ligands (Gd-BOPTA, Gd-EOB-DTPA, Figure 1) as well as preclinical formulations of liposomal and micellar paramagnetic systems and SPIOs [11].

Another approach to potential CAs for liver MRI is to target the hepatocyte-specific ASGP-R. Because this receptor is still expressed (although in reduced numbers) on hepatoma cells, it is possible to detect liver cancer metastases to other organs, such as bones. Several types of ASGP-R targeted potential MRI CAs agents have been described and tested in cells and mice. Some examples are macromolecular bioconjugates of iron oxide nanoparticulate systems, e.g., USPIOs coated with arabinogalactan (AG) (AG-USPIO) [102,103,104,105], monocrystalline iron oxide nanoparticles (MION) conjugated to the bovine plasma protein asialofetuin (ASF) (MION-ASF) [106], and, more recently, lactobionic acid (LA) and PEG-coated SPIOs to obtain LA-PEG-SPIO [107]. The specific in vivo accumulation of the CAs in a mouse liver was monitored by a greater decrease in MRI signal intensity in the presence of these agents compared to unconjugated nanoparticles [108]. These liver-specific CAs have been used to assess a range of liver diseases [108,109]. SPIOs coated with polyvinyl benzyl-O-β-d-galactopyranosyl-d-gluconamide were delivered specifically to rat liver ASGP-R on hepatocytes, as shown by negative contrast-induced in MR images [110]. Au-speckled silica-coated spherical SPIOs functionalized with the thioglycosides 2-aminoethyl-1-thio-β-d-galactopyranoside (β-d-Gal/Au/SPIO@SiO_2_ NPs) or 2-aminoethyl-1-thio-β-d-lactopyranoside (β-d-Lac/Au/SPIO@SiO_2_ NPs), with 44 nm size (obtained by TEM) and hydrodynamic diameters of about 100 nm and 120 nm, respectively (obtained by DLS), show high colloidal stability in aqueous media due to their negative zeta potential values of −42 mV and −39 mV, respectively. Their optical and magnetic properties and their in vitro targeting of the ASGP-R1 receptor overexpressed in the HepG-2 and HLE human liver cancer cell lines showed their potential for targeted dual-modal MRI/OI of hepatocellular carcinoma (HCC) [111].

A theranostic system consisting of SPIO/DOX encapsulated in the hydrophobic core of PCL–SS–GPPs micelles (PCL–SS–GPPs = amphiphilic diblock poly(3-caprolactone)-β-glycopolypeptides conjugated with galactosyl and lactosyl sugar units as ASGP-R targeting ligands) was developed. DOX and SPIOs could be efficiently transported into HepG2 tumor cells by the PCL–SS–GPPs micelles, leading to excellent MRI negative contrast enhancement [112]. Another example of a theranostic system is a pullulan stabilized SPIO conjugated with the NIR emitting dye Atto-700 (P-SPIO-AT), with a 12 nm size (obtained by TEM), and average hydrodynamic size of 80 nm (obtained by DLS), with high colloidal stability that has high relaxivities (*r*_1_ = 2.2 s^−1^ mM^−1^ and *r*_2_ = 146.91 s^−1^ mM^−1^ at 1.5 T). Accordingly, in vivo MRI of a liver fibrosis rat administered with P-SPIO-AT through the tail vein showed darkening in *T*_2w_ MRI images of the fibrotic liver regions. Optical images highlighted the same region. These two experiments underlined the ASGP-R mediated targeting of P-SPIO-AT to the fibrotic liver. A current of 400 A on an aqueous solution of 5 mg/mL of P-SPIO-AT raised the temperature above 50 °C, to facilitate effective hyperthermia [113].

Another type of targeted MRI CAs relies on macromolecular bioconjugates and polymer scaffolds as carriers bearing efficient Gd^3+^ complexes or spin labels as reporter groups and pendant β-galactoside and/or *N*-acetyl-β-galactosaminyl residues as targeting vectors to ASGP-R. Early examples of application of such macromolecular systems as positive MRI CAs are a Gd-DTPA conjugate of polylysine (PL) derivatized with galactosyl groups (Gd-DTPA-Gal-PL) [114] and a spin-labeled arabinogalactan [115]. More recent examples include Gd-DOTA macromolecular conjugates, e.g., with carboxymethyl-arabinogalactan (AG-CM) through an ethylenediamine (EDA) linkage group (Gd-DOTA-AG-CM-EDA) [114]. MRI experiments showed significant enhancement in rat liver following the intravenous administration of Gd-DOTA-AG-CM-EDA [116].

Both the particle and the macromolecular-based ASGP-R-targeted imaging agents described above include carriers bearing multiple reporter groups and a multivalent display of galactosyl targeting groups. However, these agents have the drawback of being inherently polydisperse and ill-characterized. Chemically well-defined, monodisperse and characterized multivalent agents can be assembled by an alternative molecular design: the conjugation of dendrimeric clustered carbohydrate bifunctional reagents through spacers to an MRI reporter group. A series of medium-sized Gd^3+^ complexes of DTPA- and DOTA-type ligands substituted in their periphery with one or more targeting group(s) consisting of a clustered carbohydrate of variable valence, with different topologies (Figure 7) [117] containing an increasing number of terminal galactosyl (Gal), lactosyl (Lac) or glycosyl (Glc) groups have been synthesized and studied [116,117,118,119,120]. One family of ligands included several DOTA monoamide derivatives (Figure 8), with one (DOTAGal, DOTAGlc, DOTALac), two (DOTAGal_2_, DOTAGlc_2_, DOTALac_2_) or four (DOTAGal_4_) terminal sugar groups, one DOTA *cis*-bisamide derivative with two terminal sugar groups (DO2A(*cis*)Gal_2_) and DTPA bisamides with two (DTPAGal_2_, DTPALac_2_) or four (DTPAGal_4_) terminal sugar groups. All dendrimeric sugar units were bound thioglycosi- dically, to prevent them from being cleaved off by enzymes [118,119,120,121]. Relaxometric studies showed that the relaxivity increase of these dendrimeric complexes relative to the respective parent compounds without the sugar derivatives, Gd-DOTA and Gd-DTPA-BMA, was much lower than that expected for their molecular weight increase. This was attributed to the high internal mobility of the sugar side-chains and the spacers connecting them to the more rigid part of the chelate, whose peripheral location in the conjugates did not allow an effective coupling between the Gd-OH_2_ vector and the tumbling motion of the whole complex.

Initially, these glycoconjugates were characterized pharmacokinetically by dynamic γ-scintigraphy of the [^153^Sm]^3+^-labeled analogs in vitro in human HepG2 tumor cells and their biodistributions determined in Wistar rats and mice [120,121]. The liver uptake of the labeled compounds was found to depend on their valency, sugar type, and topology, as expected from the cluster glycoside effect. For example, the affinity decreased depending on the nature of the terminal sugar groups in the order Gal > Lac > Glc. Blocking the ASGP-R in vivo by ASF reduced liver uptake by 90%, strongly suggesting that the liver uptake of these compounds is mediated by their binding to the ASGP-R receptor. However, despite the specific liver uptake of the radiolabeled galactosyl-bearing compounds, a dynamic contrast-enhanced MRI assessment of the corresponding Gd^3+^ chelates in mice showed liver-to-kidney contrast effects which were not significantly better than those shown by Gd-DTPA (Figure 9) [121]. This was rationalized by the quick wash-out of these highly hydrophilic complexes from the liver, preventing sufficient accumulation within the hepatocytes via receptor-mediated endocytosis. The cluster glycoside effect was considered also by Takahashi and co-workers, who proposed the synthesis of CAs based on a dendritic architecture containing four and twelve glucose moieties on the molecular surfaces [122].

Two medium-sized Gd^3+^ complexes with DOTA ligands symmetrically α-substituted at the four pendant acetate arms with dendrimeric sugar structures ([GdgDOTA-Glu_12_(OH_2_)]^5−^ and [GdgDOTA-Glu_12_Gly_4_(OH_2_)]^5−^), in which the Gd^3+^ ion lies at the barycenter of the macromolecular structure and, by residing upon any axis of reorientational motion, provides an effective coupling between the Gd-OH_2_ vector and the tumbling motion of the whole complex (Figure 10) [123]. The four trisaccharide dendritic wedge amine structures contain three β-glycosyl units connected, either directly ([GdgDOTA-Glu_12_(OH_2_)]^5−^) or through glycine spacers ([GdgDOTA-Glu_12_Gly_4_(OH_2_)]^5−^), to the four pendant acetate arms at α-positions (Figure 10). These larger and compact complexes have much higher relaxivities than the Gd^3+^ complexes of the monoamide DOTA derivatives described above, also resulting from the contribution of second coordination sphere water molecules hydrogen-bonded to the hydroxyl groups of the sugars.

MRI experiments at 2 T using a mouse model of a mammary tumor expressing the HER-2/neu receptor showed that [GdgDOTA-Glu_12_(OH_2_)]^5−^ caused a stronger and longer-lived signal enhancement of the tumor area than the commercial CA Gd(HPDO3A) at the same dose, with an excretion primarily via the renal system and no liver retention (Figure 11) [123]. However, in this preliminary study, the in vitro binding affinity of [GdgDOTA-Glu_12_(OH_2_)]^5−^ to the HER-2/neu receptor was not determined. A similar MRI study using the β-galactosyl analog of [GdgDOTA-Glu_12_Gly_4_(OH_2_)]^5−^ showed no significant liver signal enhancement, despite the presence of the twelve peripheral galactosyl units [123]. These results may mean that this compound, like the compounds depicted in Figure 9, is too hydrophilic to be efficiently concentrated within the hepatocytes via ASGP receptor-mediated endocytosis.

## 7. Sialic Acids as Tumor Markers

α-Linked SA, α-linked fucose, and β-linked galactose constitute the majority of monosaccharides at non-reducing terminal positions of mammalian glycans [124]. Since they are easily accessible and alterations in their expression are often associated with disease, they are attractive targets for molecular imaging. Particularly interesting are the SAs, a family of 43 monosaccharides with a 9-carbon backbone having unique structural features such as a negatively charged carboxylate under physiological conditions and an exocyclic glycerol function. In humans, the predominant member of this family of sugars is 5-*N*-acetylneuraminic acid (Neu5Ac, see Figure 12), but the corresponding 5-*N*-glycolated and *O*-acetylated neuraminic acids occur as well, particularly on tumor cells. Tumor cells often display much higher levels of sialylation than healthy cells due to increased activity of sialyltransferase [125,126,127].

HmenB1 antibodies are known to target specifically poly-SA, which is a marker of neuroblastoma, lung carcinoma, and Wilms’ tumors. These antibodies have been conjugated to rhodamine dye-doped aminated silica NPs (30 nm diameter), each of which was also linked to about 10 Fe_3_O_4_ NPs (9 nm diameter) [128]. The resulting core-satellite NPs had a high *r*_2_* (397 s^−1^ mM^−1^ at 9.4 T) and performed well as a selective targeting dual MRI/OI probe for neuroblastoma model cells. In another study, the HmenB1 antibodies were conjugated to heterodimer NPs consisting of a 6 nm FePt sphere attached to a 10 nm Au sphere [129]. This system was tested in the same cell culture and appeared to give also rise to a significant *r*_2_* increase.

The exocyclic glycerol side chain of SAs in glycans can be exploited for the binding of synthetic compounds with targeting moieties based on boronic acid. These compounds can therefore also be regarded as synthetic selectins for binding of SA (siglecs). Boronic acids can reversibly and covalently bind diol functions under the formation of five- and six-membered boronate esters [130]. The stability of these esters is mainly determined by their steric strain. The threo diol function at C_8_-C_9_ is preorganized for the formation of a five-membered boronate ester, whereas the C_7_ and C_9_-OH groups are favorably located for the formation of a six-membered boronate ester. ^11^B and ^13^C NMR, and molecular modeling studies on 2-*O*-methyl Neu5Ac as a model for α-linked Neu5Ac in a glycan have confirmed that the boronate binding predominantly occurs at these positions [131]. It should be noted that for the binding of free Neu5Ac by boronic acids at pH < 9, the geminal diol function formed by the undissociated carboxylic OH and the 2-OH groups constitutes a second favorable binding site [131]. Various boronate-based sensors have been designed for free SA [130] Conjugates of phenyl boronate with lanthanide and complexes of DTPA and DOTA have been constructed for imaging and radiotherapy of tumors having overexpression of SA on tumor cell surfaces [132,133,134,135,136]. Ammonium groups were incorporated to boost the affinity for SAs through an additional electrostatic interaction with the carboxylate groups of the Neu5Ac substrate. Accordingly, the association constant of Gd-DTPA-(EN-PBA)_2_ (see Figure 12) with free Neu5Ac (*K*_a_ = 50.4 M^−1^ at pH 7) is considerably higher than that of phenylboronic acid (PBA) with this sugar (*K*_a_ = 11.6 M^−1^) [132]. Interactions of the radioactive ^160^Tb analog with SAs on cell surfaces have been investigated using a human glioma cell line as a model for tumor tissue [133]. After incubation for 2 h, about 75% of the available ^160^Tb appeared to be present on the cells. Experiments with analogs lacking the PBA or the ammonium group showed 4–9 times less radioactivity, whereas cells from which SAs were removed by sialidase showed almost no radioactivity. It can be concluded that Tb-DTPA-(EN-PBA)_2_ interacts selectively with SA on the cell surface. However, after the binding at the cell surface, ^160^Tb^3+^ ions dissociate out of the complex and possibly move to the phosphate groups at the cell membrane. Therefore, further studies were performed with the kinetically more stable Gd-DOTA-EN-PBA [134]. To increase the relaxivity per PBA group, SA-targeted CAs with a payload of two Gd^3+^ions were developed, (Gd-DOTA-EN)_2_-PBA, and (Gd-DOTA-EN)_2_-2,3-difluoroPBA [135,136]. In vitro ^1^H NMR longitudinal relaxation rates with Gd-DOTA-EN-PBA correlated very well with the amount of SA on the surface of murine melanoma B16-F10 cells [134]. Competition by glucose resulted in only a slight reduction of the relaxation rates. In vivo MRI experiments were carried out on mice models bearing a tumor xenograft obtained by subcutaneous injection of B16-F10 melanoma cells. The obtained images (Figure 13) demonstrate that Gd-DOTA-EN-PBA can visualize the heterogeneity of the tumor much better than the commercial (non-targeting) contrast agent Gd-HPDO3A at the same Gd dose. In vivo PET imaging of B16-F10 tumor-bearing SCID mice injected intravenously with [^68^Ga]-DOTA-EN-PBA confirmed the high specificity of the tracer towards overexpressed SA. [137] It may be concluded that Gd/^68^Ga DOTA-EN-PBA has great potential for PET/MR image-guided therapy of cancer.

A CA carrying multiple Gd-DTPA groups has been fabricated by conjugating polylysine with Gd-DTPA, PBA, and rhodamine [138]. Confocal microscopy on HepG2 cancer cells showed that this agent accumulated on the cell surface; uptake was not observed. Once again, it was shown that glucose did not affect the affinity of the CA for SA.

Very high sensitivities were obtained with a bio-orthogonal chemical reporter strategy [139], using CEST between free and included hyperpolarized ^129^Xe with a probe targeted to metabolically engineered cell surfaces [140]. First azido groups were incorporated in glycan by incubation of the cells with peracylated N-azidoacetylmannosamine (Ac_4_ManNAz, Figure 14). Then, the azidosialic acid functions generated were bio-orthogonally labeled by Staudinger ligation with a reporter consisting of a peptide scaffold, to which the bioorthogonal functional group (bicyclo[6.1.0]nonyne), a ^129^Xe host (cryptophane-A), and carboxyfluorescein were attached. After loading with hyperpolarized Xe, nanomolar concentrations of SA on a cell surface could be visualized using CEST ^129^Xe MRI and fluorescence spectroscopy.

A conjugate of Gd-DTPA and PBA has been suggested as a theranostic agent for combined MRI and neutron-capturing therapy through ^10^B and ^157^Gd [141]. A platform for SA-targeted image-guided therapy was developed by dispersing Fe_3_O_4_ NPs in a synthetic nanoclay [141]. The resulting NPs were coated with PEG_5000_-PBA. The obtained material has a high *r*_2_ relaxivity (266 s^−1^mM^−1^) and appeared to be efficient as photoacoustic imaging agent. It could be used in photo thermal therapy upon NIR absorption. Another platform consisting of carbon nanotubes loaded with SPIOs, porphyrin, and a PEG_5000_ derivative with 4 PBA groups as SA-targeting functions has been employed for combined *T*_2w_ MRI, fluorescence imaging, and photodynamic therapy [142].

## 8. Sensing of Glycated HSA and Hemoglobin

In the presence of excessive levels of glucose that occur in diabetes patients, a non-enzymatic reaction with free amino groups in proteins, such as human serum albumin (HSA) and hemoglobin (Hb_A0_), takes place to form a Schiff base. The latter spontaneously undergoes an Amadori rearrangement to the protein fructosamine (HbA_1c_). This and consecutive reactions are associated with complications of diabetes, including cardiovascular diseases, retinopathy, nephropathy, and neuropathy. Blood glucose control is important for preventing and slowing down these adverse effects, but it may be useful to quantify occasionally also the extent of glycation of HSA and Hb to obtain a retrospective indicator of the average glucose concentration over the previous 1–3 weeks and 2–3 months, respectively [143].

NMR studies have proven that fructosamine in aqueous solutions occurs as an equilibrium of the β-pyranose (58%), β-furanose (19%), and α-furanose (24%) anomers [144]. The latter is perfectly pre-organized for binding with phenyl boronate (PBA) and, consequently, fructosamine is strongly and reversibly bound as PBA ester of this anomeric form, which experiences an additional stabilization by electrostatic interaction between the neighboring negatively charged boronate and the positively charged ammonium function (Figure 15). Aime et al. have exploited this for the quantification of glycated HSA with the use of the conjugate of Gd-DTPA and PBA, (Gd-DTPA-PBA_2_), see Figure 16 [145]. The binding of the boronate functions of this compound to the fructosamine residues of glycated albumin gives rise to an enhancement of the longitudinal water proton relaxation rate as a consequence of the increased *τ*_R_. A similar mechanism of interaction was observed for fructosamine model compounds with a conjugate of PBA and a lanthanide DTPA derivative, in which the central pendant arm was replaced by the methyl amide of l-lysine [146]. Unfortunately, this complex also shows a rather strong interaction with unglycated HSA, and, therefore, it is not suitable for the determination of the degree of glycation of HSA. Unexpectedly, the binding of (Gd-DTPA-PBA_2_) to fructosamine residues in oxygenated Hb_A1c_ is about equally strong as its binding to oxygenated Hb_A0_ [147], whereas the *R*_1_ enhancement with the latter is much larger. Based on extensive NMR and UV measurements, it has been proposed that the interaction with Hb_A0_ involves coordinative N–B bonds at two histidine residues of different β-chains of the protein. Steric hindrance of the access to the (Gd-DTPA-PBA_2_) binding site in Hb_A0_ explains a reduced binding strength to this site upon glycation to Hb_A1c_.

Sherry and co-workers have constructed a glucose sensor based on PBA derivatives of the Eu^3+^ complexes of DOTA-tetraamide [148,149]. The 1,7-disubstituted derivative (Eu-DOTAM-Me_2_-PBA_2_; Figure 17) has the PBA groups located in an optimal position for strong and selective binding of glucose (apparent stability constant of the resulting boronate ester: 339 M^−1^ at pH 7). Glucose is bound to Eu-DOTAM-Me_2_-PBA_2_ in a 1:1 fashion by forming a bridge above the Eu^3+^-bound water molecule resulting in a reduction of the exchange rate of that water molecule with bulk water by a factor of two. The residence time of the Eu^3+^-bound water molecule then is sufficiently slow on the NMR time scale to permit presaturation of the ^1^H resonance of the Eu^3+^-bound water resonance to modulate the intensity of the bulk water through PARACEST and thus, the MRI contrast, which can be exploited for mapping the distribution of glucose in tissues. This has been demonstrated by PARACEST images of livers perfused in the magnet [150] which responded to the alterations of glucose concentrations upon stimulation of glycogenolysis by the hormone glucagon.

The binding strength of Eu-DOTAM-Me_2_-PBA_2_ to glycated HSA is of the same order of magnitude as that of glucose. Since the plasma concentration of glycated HSA is normally relatively low, this should not interfere with the use of the complex as a sensor for glucose in tissue. On the other hand, the good affinity for glycated HSA allows the in vitro determination of the degree of glycation in serum, using MRI and high-throughput methods after the separation of excess simple sugars [149]. More recently, a modified sensor has been developed, in which the uncharged methyl substituents on the 4,7-amide functions were replaced by CH_2_COO^−^ groups [151]. In contrast to the positively charged Eu-DOTAM-Me_2_-PBA_2_, this agent exhibited no clinical signs of toxicity in mice, whereas it had similar sensitivity to glucose.

## 9. MRI Contrast Agents with Glucose or Derivatives as a Targeting Group

Glucose (Glc) is the most important source of energy for cells and consequently, alterations in glucose uptake and metabolism are important biomarkers of malfunction due to diseases such as cancer, cardiovascular problems, Alzheimer’s disease, and neurologic and psychiatric disorders. Cells in many tumors meet their large energy demand, among other things, by upregulation of the Glc transporting transcellular membrane receptors (particularly GLUT-1 and GLUT-3) and increased anaerobic glycolysis (the Warburg effect) [152], resulting in increased lactic acid production and an acidic extravascular and extracellular space. This is exploited in PET with the use of 2-deoxy-2-[^18^F]fluoro-d-glucose (FDG) as a tracer [153]. GLUT receptors also transport FDG into the cell, where it is phosphorylated, but then the F-atom at the 2-position prevents further conversions and consequently, FDG is trapped in the cells, which can be visualized by PET. Since the radioactivity of FDG limits the scan frequency excluding certain patient groups and because the relatively short *t*_½_ of ^18^F (109.77 min) can give rise to logistic problems, alternatives are in demand. Since 2-deoxyglucose (2-DG) and many of its derivates are also accumulating in tumor cells, conjugates of 2-amino-2-DG and DTPA have been used as *T*_1_ CAs to detect metabolically active tumor tissues in cancer cell lines and xenograft tumor models [154,155].

The low concentration of both Glc in plasma and of Glc receptors combined with the low sensitivity of MRI usually requires that MRI alternatives of FDG apply magnification techniques to obtain sufficient sensitivity for detection. For instance, the clinical CA Gd-EOB-DTPA has been entrapped in the cavity of spherical non-ionic vesicles (niosomes) with the targeting vector N-palmitoyl glucosamine (NPG) included in the membrane [156]. With human prostate adenocarcinoma PC3 cells implanted in mice, it was found that the combination of NPG and PEGylation led to an increased tumor/muscle contrast to noise ratio, probably due to targeting over-expressed GLUT receptors by the NPG groups on the surface of the niosomes.

A *T*_2_ CA has been constructed by coating USPIOs (γ-Fe_2_O_3_) with 2,3-dimercaptosuccinic acid (DMSA) followed by amidation with 2-amino-2-DG [157,158]. The resulting γ-Fe_2_O_3_@DMSA@DG NPs had a diameter of 10 nm. Prussian blue staining, TEM, and inhibition by competition with the antibody of GLUT-1 showed a high and selective uptake by several GLUT-1 overexpressing breast tumor cell lines (MDA-MB-231, MCF7, and ZR-75-1). The endocytosis was also reflected by a substantial decrease in MRI intensity, which was significant compared to that of breast fibroblasts. [158] Similarly, USPIOs that were first coated with 3-aminopropyltriethoxysilane and then amidated with 2-amino-2-DG have been employed to demonstrate the targeted uptake in U-251 human glioma cells with expression of GLUT-1 and GLUT-3 [159].

## 10. Enzyme-Responsive Contrast Agents Containing a Carbohydrate Group

Carbohydrates are ubiquitous and have a wide array of biological roles. Since many chemical conversions involve enzymes, diseases are often reflected in alterations of certain enzyme activities. Therefore, probes that are capable of responding to enzyme activity have great potential in the diagnosis and monitoring of the effects of therapy [14,160,161]. In the field of glycobiology, the main enzymes of interest are galactosidase, glucuronidase, and hyaluronidase.

The first example of an enzyme-activated MRI CA reported was Gd-HE-DO3A conjugated to a β-d-galactopyranose (Gd-HE-DO3A-Gal, Figure 18) [162]. The β-galactose group of Gd-HE-DO3A-Gal is limiting the access to the inner coordination sphere of Gd^3+^ for water molecules. After the removal of the sugar by the enzyme β-galactosidase, an OH-group remains at Gd-HE-DO3A and coordinates to Gd^3+^. The less crowded geometry of the complex allows increased accessibility of water into the first coordination sphere of Gd^3+^, resulting in a 25% increase in *r*_1_ (at 11.7 T and 37 °C). Since this results in insufficient contrast for adequate in vivo employment, an α-methyl group was attached to the pendant arm of the chelate to give Gd-HP-DO3A-Gal [163,164]. Because of the chirality of the C-atom, which carries the methyl substituent, this complex occurs as two diastereomers [165]. Luminescence and relaxivity studies indicated that both diastereomers have almost no water in the first coordination sphere of Gd^3+^ (*q* ≈ 0). In one of these isomers, the galactose unit is bent away from the Gd-DO3A residue and consequently, there is ample space for binding of endogenous carbonate in a bidentate fashion (Figure 19A). In the other diastereomer, the galactose unit is located over the Gd^3+^ ion and blocks access of water molecules into the first coordination sphere (Figure 19B). Cleavage of the sugar removes the shielding of the Gd^3+^ ion, whereas the hydroxyl group that remains on the appending arm coordinates Gd^3+^, expelling the carbonate in isomer A. Both cleavage products have *q* = 1–2, resulting in a 40–50% increase in relaxivity with respect to Gd-HP-DO3A-Gal (*q* ≈ 0), which provided sufficient MRI contrast for in vitro and in vivo visualization and localization of β-galactosidase, a marker of gene expression in *Xenopus laevis* embryos. This approach was further explored in the development of in vitro assays for the determination of the activity of β-glucuronidase [166], an enzyme with an enhanced extracellular concentration in necrotic areas of tumors. For this purpose, a Gd-DO3A chelate was designed that was conjugated with a β-glucuronic acid moiety, via a self-immolative nitrodihydroxybenzyl linker (Gd-DO3A-GlcA, Figure 18). Enzymatic hydrolysis of β-glucuronic acid triggers a cascade reaction that releases the Gd-chelate of 2-aminoethyl-DO3A (Gd-DO3A-AE), the bridging arm, and CO_2_. An increase in *r*_1_ of 17% was observed when Gd-DO3A-GlcA was treated with β-glucuronidase in a buffer mimicking in vivo anion concentrations. Unfortunately, *r*_1_ decreased by 27% for the same experiment in human blood serum. Apparently, interactions of the Gd^3+^ complexes with endogenous compounds, such as carbonate, are interfering with glucuronidase sensing.

Higher changes in enzyme-induced relaxivity have been obtained with the β-galactosidase responsive CA Gd-DOTA-FP-Gal (Figure 18) [167]. Upon galactose cleavage by the enzyme, an electrophilic intermediate is formed, to which proteins such as the enzyme or HSA bind covalently. The resulting large increase in *τ*_R_ gives rise to an observable increase of *r*_1_ (about 60% at 0.47 T and 37 °C). In vivo studies showed a high-intensity enhancement in mice with an implanted CT26/β-gal tumor with β-galactosidase gene expression but not for the CT26 tumor without β-galactosidase gene expression. Hanaoka et al. demonstrated that enhancement of similar magnitude can be obtained with Gd-DTPA bound to β-galactose via a biphenyl group as a linker [168]. Removal of the masking galactose group exposes the biphenyl group, which binds non-covalently with HSA, resulting in a 57% increase in *r*_1_ due to the slowing down of the molecular tumbling.

These relaxivity enhancement methods might be hampered in cellular and animal applications, because β-galactosidase resides in the cytoplasm, whereas HSA occurs mainly extracellularly. To overcome this, cell-penetrating probes have been designed. Arena et al. have constructed a probe that is capable of reporting the gene expression of β-galactosidase in melanoma cells by connecting DOTA-amide to tyrosine-β-d-galactopyranose (DOTA-Tyr-Gal, Figure 20) [169]. After the removal of galactose by β-galactosidase, the tyrosine group oligomerizes under the influence of tyrosinase which is over-expressed in melanoma cells. The resulting oligomer accumulates in the cytoplasm leading to an increase in *τ*_R_ from 0.1 to 5.6 ns, which is reflected in a 3-fold increase in *r*_1_ at magnetic field strengths between 0.5 and 1.5 T. This system has been successfully tested in B16−F10*LacZ* transfected cells and on murine melanoma tumor-bearing mice.

Tóth and co-workers have connected an Ln-DOTA unit through a benzylocarbamate self-immolative linker to β-d-galactopyranoside (Ln-DOTA-Gal, Figure 21) [170,171]. The Gd^3+^ complex showed only a modest decrease in *r*_1_ (10–20%) due to the decrease in *τ*_R_ upon its decomposition under the influence of galactosidase, and therefore, it is not suitable as responsive *T*_1_ CA. However, the corresponding Yb^3+^-complex (Yb-DOTA-Gal) appeared to be a very effective PARACEST CA. This compound did not show a CEST effect, although it has an exchangeable carbamate proton, but upon attack by β-galactosidase, an electron cascade initiates the cleavage of carbamate, followed by transformation into an amine (Yb-DOTA-NH_2_). The slowly exchanging magnetically nonequivalent amine protons were used to generate a PARACEST effect with a magnitude that is dependent on the pH. Upon protonation, the exchange rate of the amine protons increases, which is reflected in an enhancement in the CEST effect. This novel class of CAs has great potential for the detection of a variety of enzymes by a simple change of the glycol-substrate. The same group developed a set of three enzyme-responsive lanthanides (Gd^3+^, Tb^3+^, Yb^3+^) complexes of the ligand DO3A-pyMe-carb-Gal (Figure 21) that that can be monitored independently in three complementary imaging modalities [172]. The complexes are almost isostructural. Upon β-galactosidase cleavage, the produced Ln^3+^ complexes differed from the initial Ln^3+^ complexes in the hydration state (*q* = 0 → *q* = 1), which is reflected in an increase in *r*_1_ for the Gd^3+^ complexes (*r*_1_ = 1.91 → 3.77 s^−1^ mM^−1^ at 1.4 T, 25 °C) and switching off of the luminescence for the Tb^3+^ complexes. A single CEST peak was observed for the carbamate proton of Yb-DO3A-pyMe-carb-Gal, which disappeared after enzymatic cleavage.

Recently, diamagnetic self-immolative CAs were developed that are capable to quantify the activity of β-galactosidase and β-glucuronidase [173]. These agents consisted of a glycosyl substrate (β-galactose or β-glucuronic acid) and a salicylic acid moiety connected by a nitrobenzyloxy-carbamate spacer (Figure 22). The ^1^H NMR spectrum displays CEST signals for the carbamate proton (4.25 ppm) and the salicyl carboxylate proton (9.25 ppm). The exchange between these protons and water protons is suitable for CEST, but the exchange among carbamate and salicylate protons is too slow for mutual saturation. After enzyme-mediated cleavage of the CA, the signal at 9.25 ppm remains as an unresponsive CEST signal, whereas that at 4.25 ppm disappears due to the conversion of the carbamate group into an amine, which allows quantification of the enzyme activity. The CA has the additional advantage that the nitrobenzyloxy group is UV-vis active.

The commercially available histologic stain 3,4-cyclohexenoesculetin-β-d-galactopyranoside forms a black precipitate in the presence of Fe^3+^ ions, after cleaving off the β-d-galactopyranoside by β-galactosidase. The paramagnetic precipitate can be detected by *T*_2w_* MRI, which may be exploited for the detection of *lacZ* gene activity, particularly at higher magnetic field strengths [174,175]. A series of dihydroxycoumarins of which one or both of the OH groups were attached to β-d-galactopyranoside exhibited some potential both as *T*_1_ and *T*_2_ galactosidase responsive CA, as demonstrated with various cells (human MCF7 breast and PC3 prostate cancer), as well as stably transfected clones expressing β-galactosidase (MCF7-*lacZ* and PC3-*lacZ*) [176].

^19^F MRI has the advantage of the absence of intrinsic background signals. β-galactosidase responsive probes have been designed, that are based on enzyme-mediated changes in ^19^F chemical shifts of β-d-galactopyranose of which the 1-OH group is derivatized with fluorinated groups [177,178,179,180]. Prototypes of dual ^1^H/^19^F galactosidase selective probes (Gd-DFP-Gal, Gd-DOMF-Gal, Figure 22) have been constructed, which exhibit self-immolative cleavage of an F-containing group following the enzyme-mediated removal of β-d-galactopyranose [181,182]. Before cleavage, the ^19^F resonance is quenched due to broadening by Gd^3+^ in its proximity. After galactose-mediated cleavage, the ^19^F signal switches on, while the ^1^H signal dims due to a decrease in *r*_1_ as a result of an increase in *τ*_R_ of the produced Gd^3+^ complex with reduced molecular volume.

Hyaluronidase degrades high-molecular-weight HA to lower molecular-weight fragments. High-molecular-weight HA is anti-angiogenic, whereas low-molecular-weight fragments can induce angiogenesis. Hyaluronidase activity correlates with aggressiveness and invasiveness of ovarian cancer metastasis and with tumor angiogenesis. A hyaluronidase-responsive CA has been designed that consisted of agarose beads connected with HA-EN-DTPA-Gd through an avidin-biotin linker [183,184]. The high-molecular-weight HA in this material could serve as a substrate analog for hyaluronidase. The resulting fragmentation of the HA was reflected in an increase of *r*_1_ and *r*_2_, most likely due to the accompanying increased exposure of Gd-DTPA to water. This behavior was demonstrated on ovarian carcinoma ES-2 cells and with corresponding mice xenografts. Micelles of high-molecular-weight HA linked through tetraethylene glycol to cholesterol have been investigated for the hyaluronidase-induced delivery of DOX and iron oxide NPs to cancer cells (HeLa, HepG2, and MCF7) [185].

The *T*_2_ CA Feridex (iron oxide coated with dextran) is responsive to the enzyme dextranase, which is capable of removing the coating resulting in a 35–40% increase in *r*_2_ and *r*_2_* (at 4 T) [186]. The *r*_2_* changes were larger in vivo than in vitro, probably due to the coagulation of the naked iron oxide NPs after the enzymatic cleavage.

## 11. Outlook

During the last decades, much progress has been made in the development of MRI contrast agents for the imaging of biomarkers of various diseases. The paramagnetic metal ions, particularly Ln^3+^ ions, play a crucial role in many diagnostic imaging techniques. Much insight has been obtained into the relationship between molecular structure and parameters governing the MRI images and many ways of increasing the sensitivity of CAs have been explored. In the field of glycobiology, a myriad of synthetic procedures for efficient MRI CAs has been developed. For many of the novel potential CAs, initial toxicity studies have been performed using cell lines and/or in vivo animal experiments. However, the barriers to the clinical translation of potential CAs are huge. The main problems are the recent concerns about the toxicity of Gd-based CAs and the commercial unattractiveness of highly specific diagnostic and theranostic agents.

Gd^3+^-based CAs are now in use for almost 30 years and have proven to be generally extremely safe. Only 0.03% of all administrations (about 100 million worldwide) gave rise to serious adverse effects. Important factors concerning safety are the high thermodynamic and kinetic stability of the Gd-chelates used and their rapid excretion. However, reports of nephrogenic systemic fibrosis (NSF), late-stage renal failure associated with Gd-based MRI contrast agents and the observation of Gd accumulation in the brains (globus pallidus and dentate nucleus) of patients with normal renal function, after repeated administrations of DTPA-type of CAs, has given rise to concerns regarding this class of compounds [187,188,189,190,191,192]. This has prompted the European Medicines Agency (EMA) to recommend suspension or restriction of the authorization of Gd-complexes of DTPA derivates (see Figure 1) [193]. However, the United States Food and Drug Administration (FDA) did not identify evidence to date that gadolinium retention in the brain from any of the gadolinium-based CA, including those associated with higher retention of gadolinium, is harmful, and that, therefore, restricting their use is not warranted [194]. Although the clinical significance of Gd^3+^ deposits in the brain is unknown, it is undesirable and it may be expected that more acyclic CAs will be banned in the near future. Many proposed CAs described in this review are of this class, but in most cases, it will be easy to substitute the DTPA framework with a kinetically more stable DOTA analog or with Gd-picienol (see Figure 1), which was introduced in the market very recently [195].

Gd-based MRI agents must be cleared from the body after contrast-enhanced MRI within a short period after administration. Prolonged residence times increase the risk of leaching toxic metal ions. Nanoparticulate CAs are under heavy scrutiny for application as more sensitive and selective CAs, but NPs have usually relatively long residence times in the body and therefore, they might be less safe. Thorough investigations of the long-term stability of NPs are missing in almost all studies. Since the residence time and the biodistribution of NPs are strongly related to the particle size, it may be expected that the smallest NPs (<10 nm diameter) are the safest.

The concerns regarding the toxicity of Gd^3+^ have initiated an increased interest in research on alternative paramagnetic metals, such as Mn^2+^ in the form of small complexes [196] and Mn-oxide NPs [197], as well as iron based CAs in the form of small high spin Fe^3+^ complexes [198,199] and NPs [200] and iron oxides and their ferrites [201,202,203] or metal-free techniques like glucoCEST [7]. However, the relaxivity and stabilities of Mn^2+^ and high spin Fe^3+^ complexes are usually inferior to that of the Gd^3+^ complexes and large doses of Mn^2+^ ions are causing irreversible neurological Parkinson-like disorders [204]. Some controversy exists about the toxicity of iron oxide NPs as well. Slowed clearance may for example give rise to reactive oxygen species mediated toxicity [205,206,207,208]. Moreover, CAs based on almost all iron oxide NPs are withdrawn from the market, probably for commercial reasons. GlucoCEST is limited by the low sensitivity of the technique [7]. For the time being, it may be concluded that systems based on Gd^3+^-complexes of DOTA derivatives or iron oxide and metal-free CAs are the most attractive as far as safety is concerned. For all CAs it is important to maximize their efficacy so that the applied doses can be minimized.

After the first introduction of Gd-based CAs, not many new clinical MRI CAs came to the market and many of those that did were withdrawn after some time, often for economic reasons. A major problem of MRI CAs is that the development costs, mainly the costs of phase III studies, are much too large in comparison with the post-approval revenues to make CAs sustainable [209,210]. It may be expected, however, that the wealth of information that can be obtained with molecular imaging using CAs as described in the present review may, on the long term, lead to early diagnosis, and effective personalized therapies, which will improve the quality of life of patients and which ultimately will result in a net reduction of the health care costs of society.

## Figures and Tables

**Figure 1 molecules-27-08297-f001:**
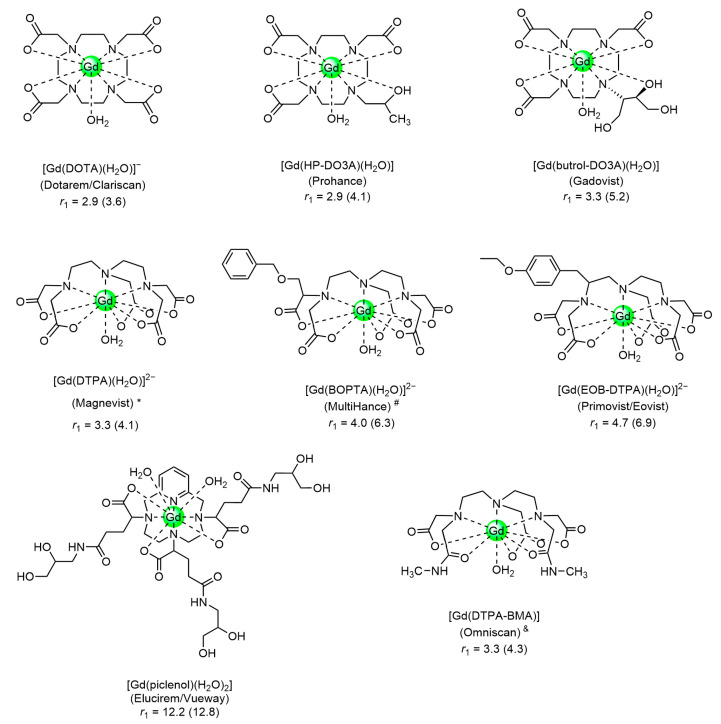
Schematic representation of the molecular structure of commercial Gd-based contrast agents. Relaxivities are in s^−1^ mM^−1^ at 1.5 T and 310 K in aqueous solutions and in biological medium (in brackets). Charges are omitted for clarity. The brand names are given between brackets. * The European Medicines Agency (EMA) recommends restricting the use of this CA only for intra-articular use. # The EMA allows this CA only for liver scans. & Authorization suspended by the EMA.

**Figure 2 molecules-27-08297-f002:**
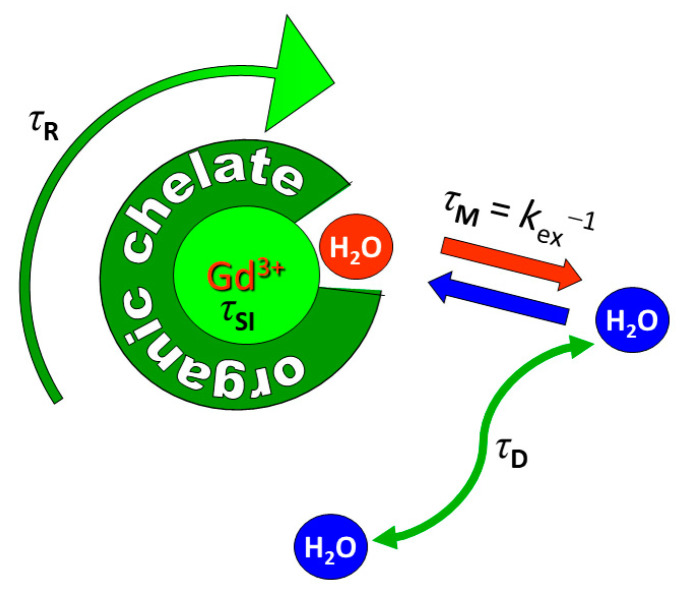
Schematic representation of the main parameters that govern the longitudinal relaxivity of a Gd^3+^ complex. Copied with permission from ref. [2]. Copyright 2016, Elsevier.

**Figure 3 molecules-27-08297-f003:**
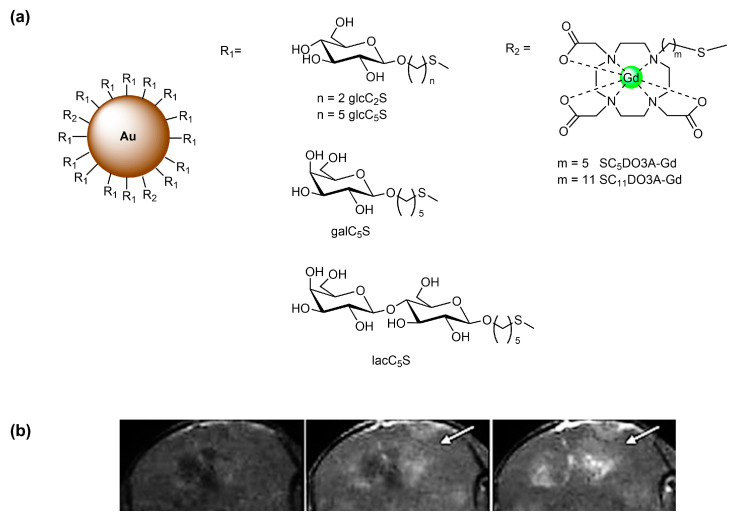
(**a**) Thiol-ending sugar conjugates and the corresponding paramagnetic glyconanoparticles; Gd-coordinated water molecules have been omitted for clarity. (**b**) Left: *T*_1w_ image of a GL261 generated tumor; Middle: *T*_1w_ image after injection of Magnevist (0.1 mmol kg^−1^, Gd(III)); Right: *T*_1w_ image after injection of glcC_5_S-Au- SC11DO3A-Gd (0.1 mmol kg^−1^, Gd(III)). The arrows indicate the tumoral zones. The MR images were acquired at 7 T. Part (**b**) copied with permission from ref. [51]. Copyright 2013, The Royal Society of Chemistry.

**Figure 4 molecules-27-08297-f004:**
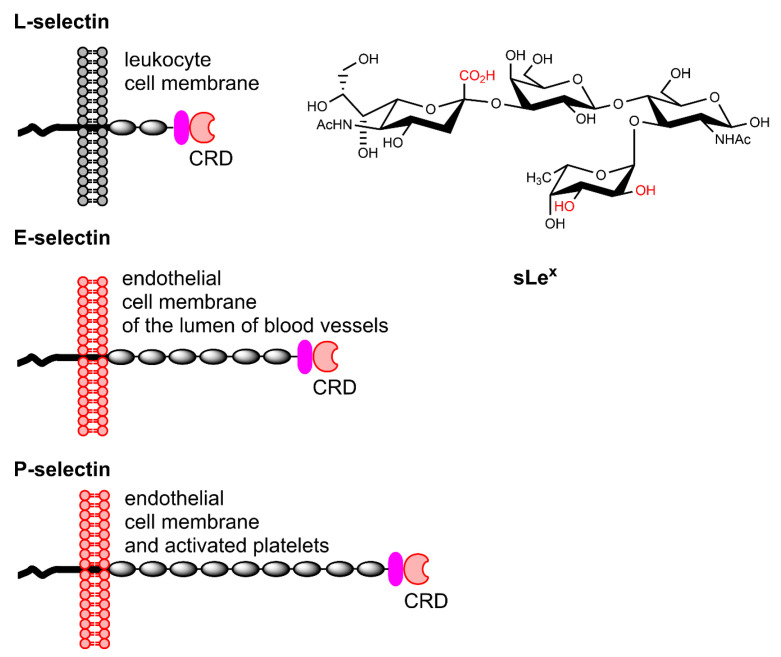
Schematic representation of the members of the selectin family, together with the molecular structure of their main ligand sLe^x^.

**Figure 5 molecules-27-08297-f005:**
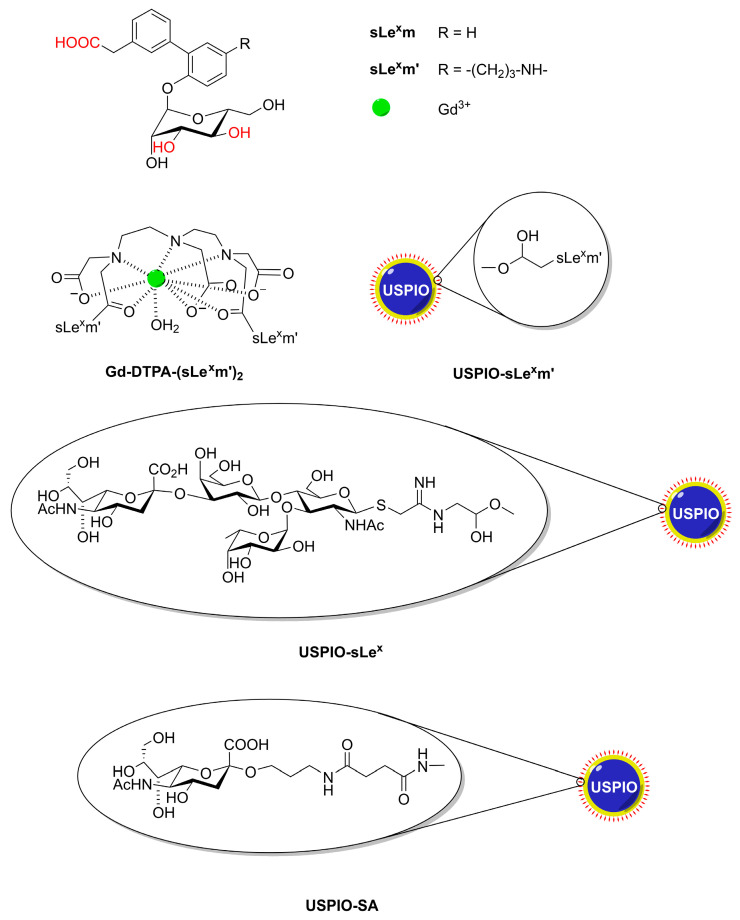
Structures of selectin targeting MRI CAs.

**Figure 6 molecules-27-08297-f006:**
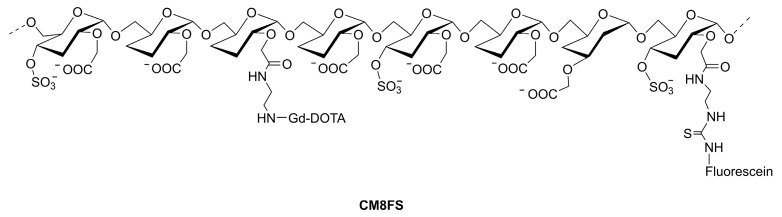
P-selectin glycoprotein ligand-1 mimetic CM8FS. Unsubstituted OH functions of the dextran backbone are omitted for clarity.

**Figure 7 molecules-27-08297-f007:**
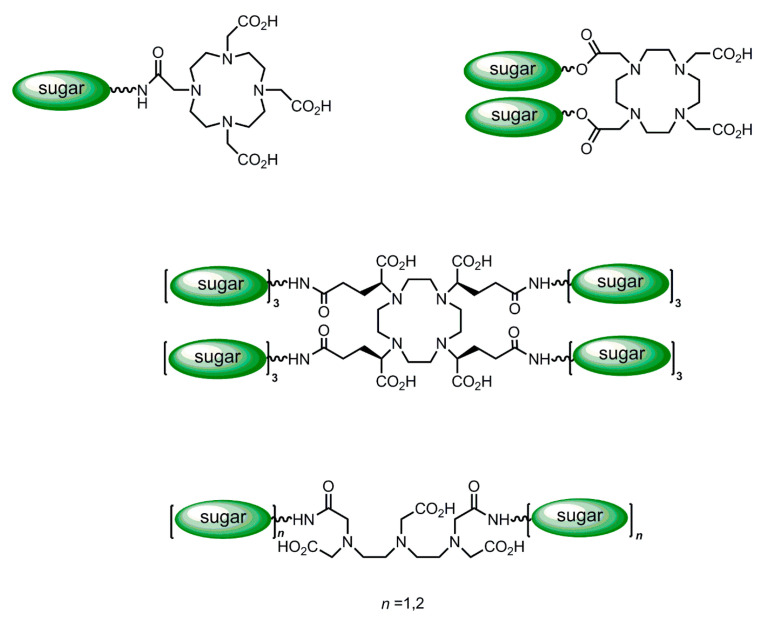
Comparison of some general topologies of glycoconjugates of DOTA (**top and middle rows**) and DTPA (**bottom row**) ligands (adapted with permission from Ref. [117]. Copyright 2009 Wiley-VCH Verlag GmbH&Co. KGaA).

**Figure 8 molecules-27-08297-f008:**
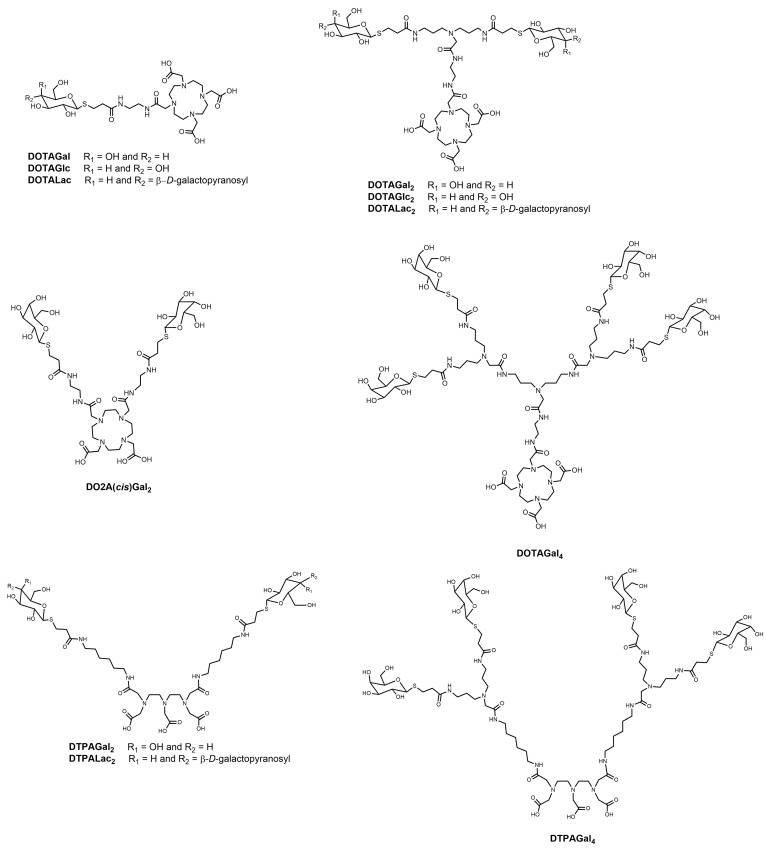
Schematic representation of the molecular structures of glycoconjugated DOTA and DTPA amide derivative ligands used as Gd^3+^ complexes in MRI CAs studies.

**Figure 9 molecules-27-08297-f009:**
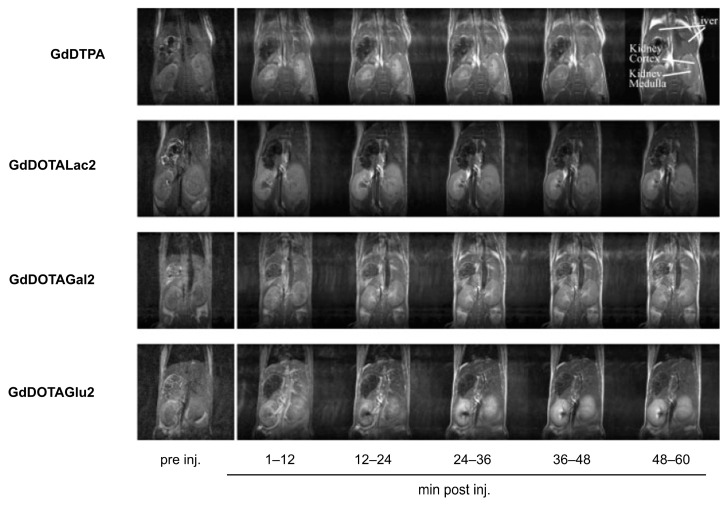
Axial *T*_1w_ spin-echo MR images of mice before and after injection of GdDTPA (dose 0.2 mmol kg^−1^ BW), GdDOTALac_2_, GdDOTAGal_2_ and GdDOTAGlc_2_ (dose 0.3 mmol kg^−1^ BW). Each image corresponds to the average of 12 min acquisition. Reproduced with permission from Ref. [121]. Copyright 2006, John Wiley & Sons, Ltd.

**Figure 10 molecules-27-08297-f010:**
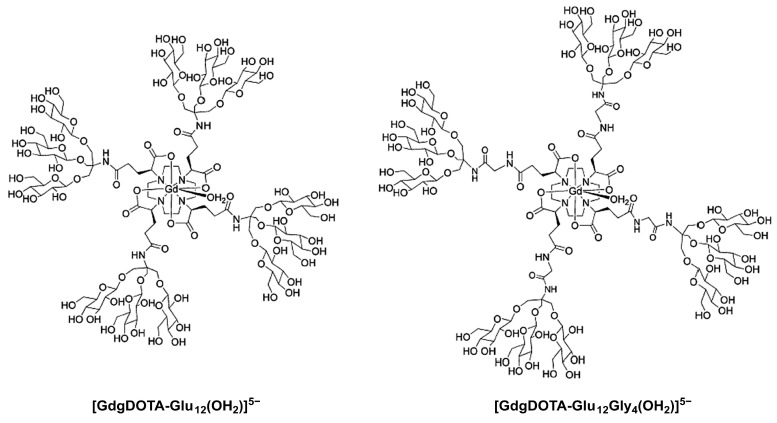
Two Gd(III) complexes of DOTA derivatives α-substituted at the four pendant acetate arms with dendrimeric sugar structures ([GdgDOTA-Glu_12_(OH_2_)]^5−^ and [GdgDOTA-Glu_12_Gly_4_(OH_2_)]^5−^) with optimal *τ*_R_ values, thanks to the location of the Gd^3+^ ion on the barycenter of their macromolecular structure.

**Figure 11 molecules-27-08297-f011:**
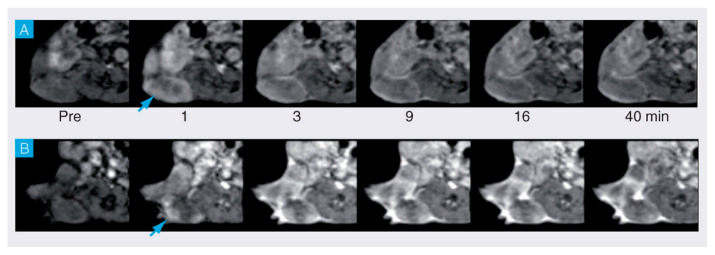
In vivo *T*_1w_ MR images acquired in a mouse model of mammary carcinoma (BALB-neuT female mouse over-expressing the transforming activated rat HER-2/neu oncogene under the control of the mouse mammary tumor virus promoter) pre- and post- administration of Gd(HPDO3A) (**A**) or [GdgDOTA-Glu_12_Gly_4_(OH_2_)]^5−^ (**B**) at the same gadolinium dose (0.1 mmol kg^−1^). The tumor signal enhancement (arrow) was two times higher with [GdgDOTA-Glu_12_Gly_4_(OH_2_)]^5−^ than with Gd(HPDO3A), further highlighting tumor structures. The contrast caused by [GdgDOTA-Glu_12_Gly_4_(OH_2_)]^5−^ showed also a slow wash-out (unpublished results provided by the authors of Ref. [123]).

**Figure 12 molecules-27-08297-f012:**
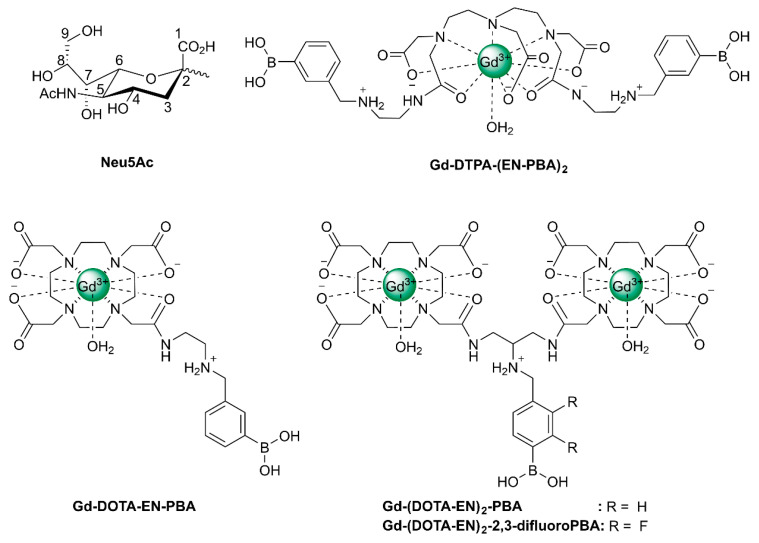
Molecular structures of Neu5Ac end group in glycoconjugates, Gd-DTPA-(EN-PBA)_2_, Gd-DOTA-EN-PBA, Gd-(DOTA-EN)_2_-PBA, and Gd-(DOTA-EN)_2_-2,3-difluoroPBA.

**Figure 13 molecules-27-08297-f013:**
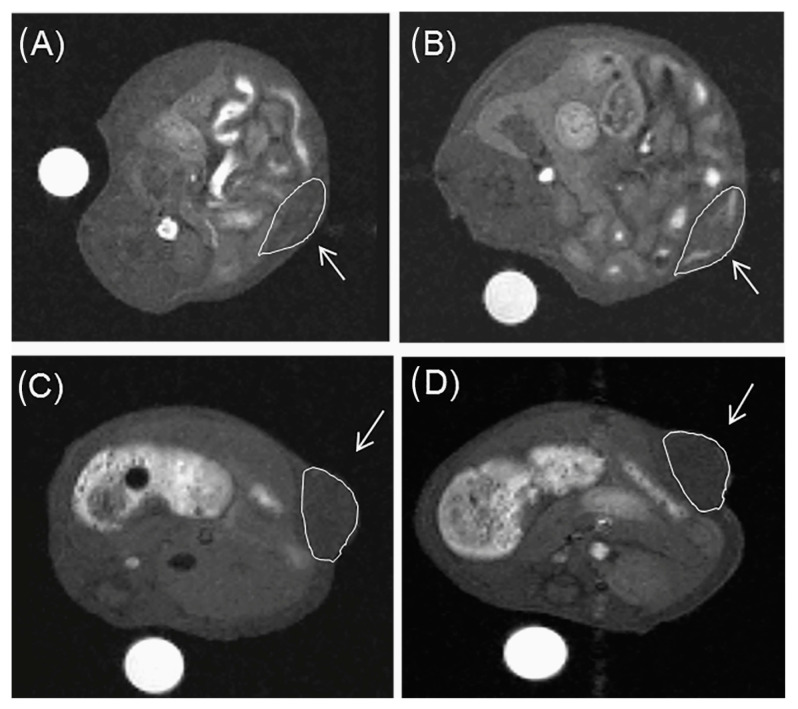
Fat-suppressed *T*_1w_ MR spin-echo images of C57BL/6 mice grafted subcutaneously with B16-F10 melanoma cells recorded at 7 T before (**A**) and 4 h after (**B**) the administration of Gd-DOTA-EN-PBA. For comparison, analogous measurements were performed before (**C**) and 4 h after (**D**) administration of Gd-HPDO3A. Copied with permission from Ref. [134] Copyright 2012 Wiley-VCH Verlag GmbH & Co. KGaA.

**Figure 14 molecules-27-08297-f014:**
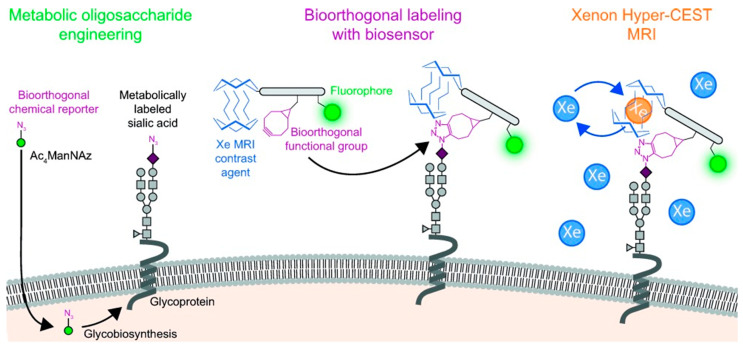
Xenon Hyper-CEST biosensors targeted to metabolically labeled glycans using bioorthogonal chemistry. Three key steps were required to image metabolically labeled glycans on live cells with xenon Hyper-CEST biosensors. Metabolic oligosaccharide engineering (**left**): cells are treated with Ac_4_ManNAz, a synthetic sugar bearing a bioorthogonal azide group, which is subsequently incorporated into the glycome as a terminal SA. Bioorthogonal labeling with the biosensor (**center**): cells are labeled with the multimodal (xenon MRI/fluorescence) biosensor bearing a complimentary bioorthogonal functional group, xenon host, and fluorophore. Xenon Hyper-CEST MRI (**right**): the final step, in which hyperpolarized xenon is delivered to the sample immediately before measurement. Xenon Hyper-CEST MRI uses the reversible binding of xenon to the host to greatly amplify the biosensor signal. Copied from Ref. [140]. Copyright 2015 Wiley-VCH Verlag GmbH & Co. KGaA.

**Figure 15 molecules-27-08297-f015:**
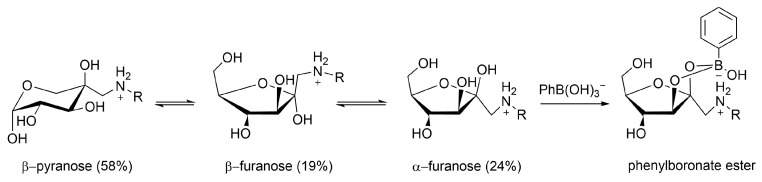
Schematic representation of the anomeric equilibrium of the fructosamine residue in glycated HSA and its phenylboronate ester. R = HSA.

**Figure 16 molecules-27-08297-f016:**
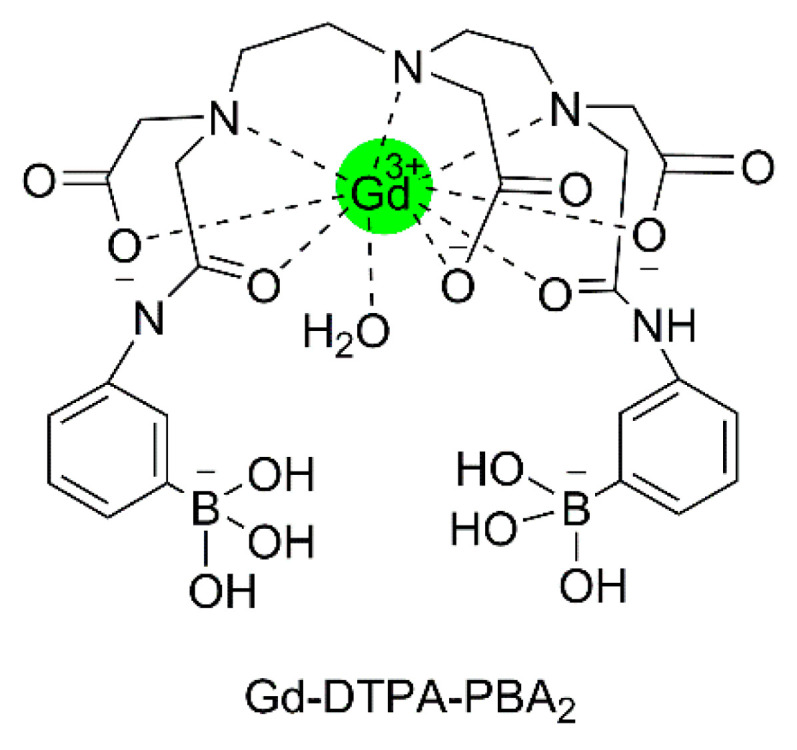
Molecular structure of Gd-DTPA-PBA_2_.

**Figure 17 molecules-27-08297-f017:**
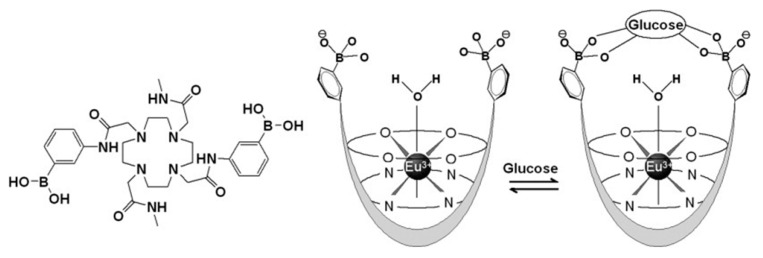
The chemical structure (**left**) and proposed binding model (**right**) for EuDOTAM-Me_2_-PBA_2_ with glucose. Copied with permission from Ref. [150]. Copyright 2008, Wiley-Liss, Inc.

**Figure 18 molecules-27-08297-f018:**
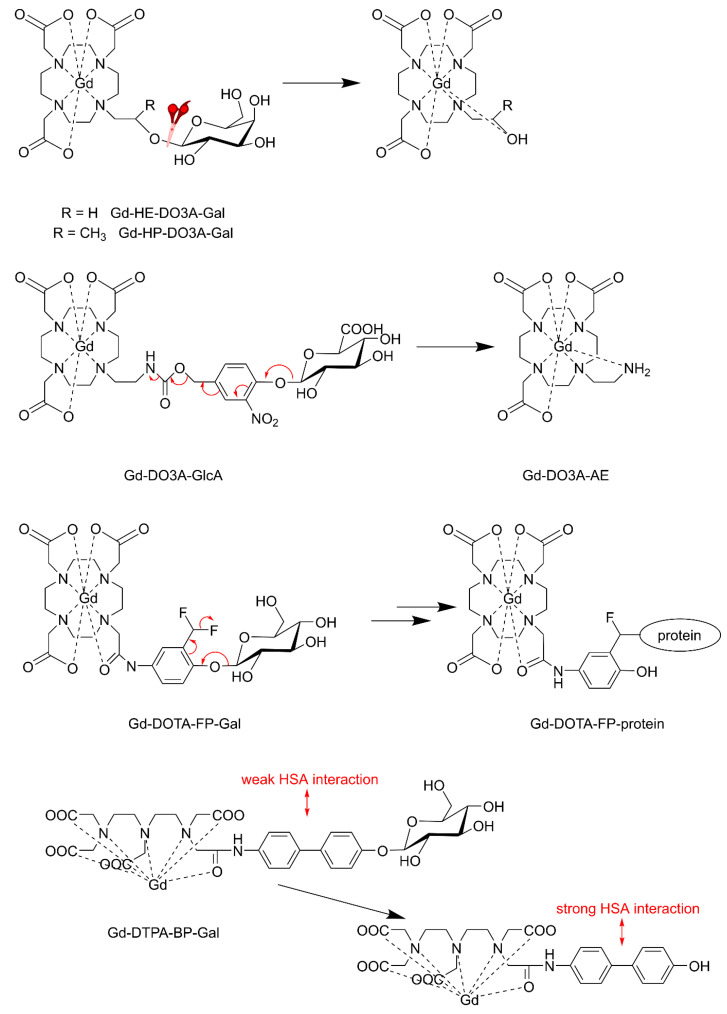
Schematic representation of the molecular structures of some enzyme-responsive CAs. Charges and Ln-coordinated water molecules are omitted for clarity.

**Figure 19 molecules-27-08297-f019:**
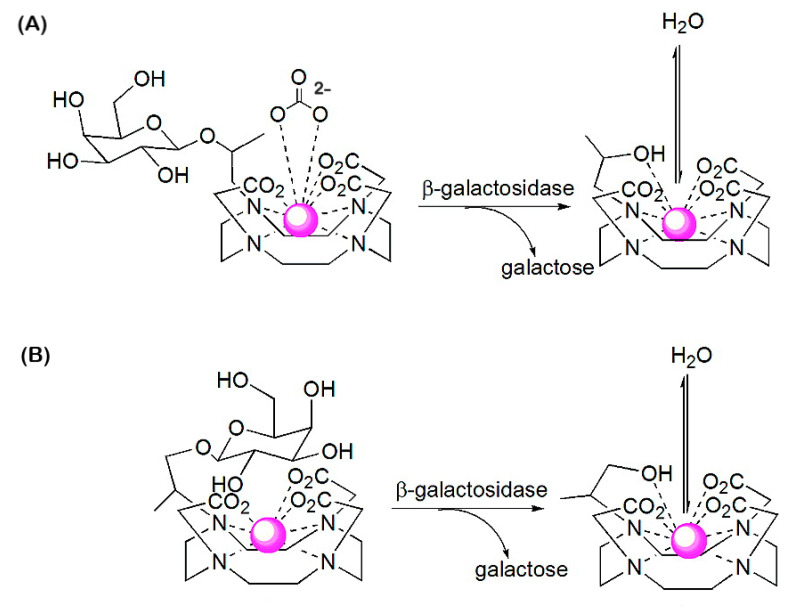
The different pathways of the β-galactosidase hydrolysis of the two diastereomers of Gd-HPDO3A-Gal. (**A**) represents the pathway for the diastereomer that has the galactose unit bent away from the DO3A moiety and (**B**) that for the other isomer with the galactose unit located above the DO3A moiety.

**Figure 20 molecules-27-08297-f020:**
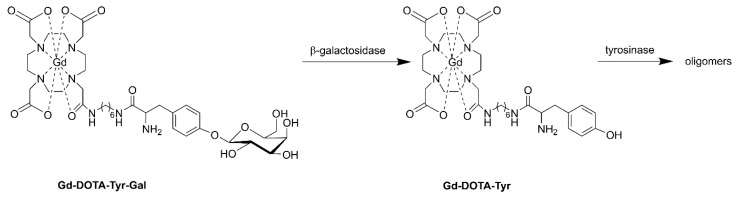
Schematic representation of the molecular structures of a galactosidase-tyrosinase responsive CA. Charges and Gd-coordinated water molecules are omitted for clarity.

**Figure 21 molecules-27-08297-f021:**
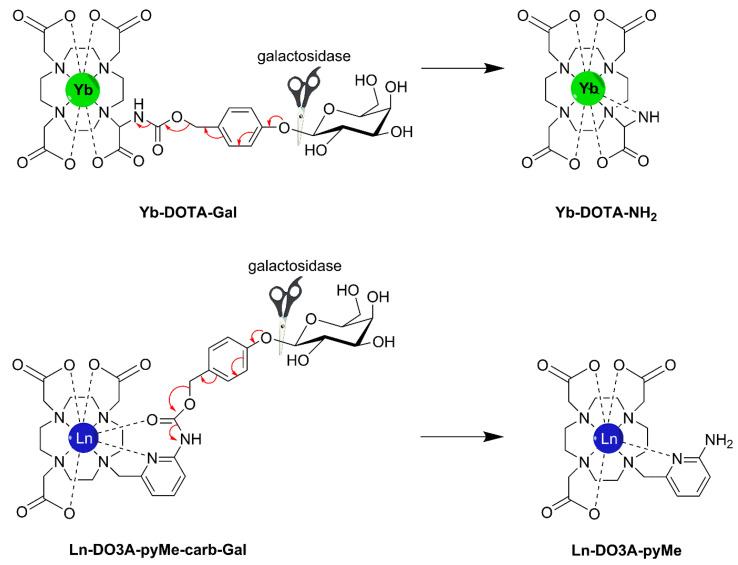
Schematic representation of the molecular structures of some β-galactosidase responsive self-immolative CAs and their Ln^3+^-containing cleavage products. Charges and Ln-coordinated water molecules are omitted for clarity.

**Figure 22 molecules-27-08297-f022:**
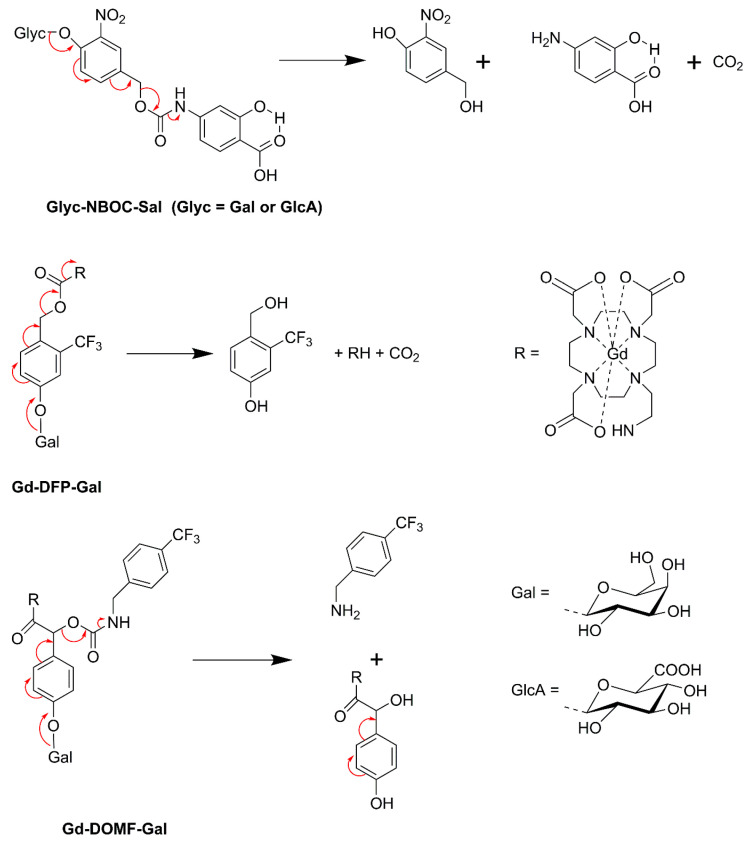
Schematic representation of the molecular structures of β-galactosidase and glucuronidase responsive self-immolative CAs and their cleavage products. Charges and Ln-coordinated water molecules are omitted for clarity.

## Data Availability

Not applicable.

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
