# Peer review of "MRI Contrast Agents in Glycobiology"

_molecules, 2022, doi:10.3390/molecules27238297_

Round 1

Reviewer 1 Report

The article is mainly a catalog of recent MRI contrast agents. While this is of interest the manuscript would greatly improve if the author would critically comment and discuss drawbacks and advantages of different conjugates and chemistries in some more detail. A multitude of biological activities and potential applications are mentioned and the manuscript would greatly improve if a handfull representative examples would be described in more detail. Over time numerous glyco-MRI agents were synthesized, but few clinical application of diagnostics exist. A critical discussion of the therapeutic potential would add value to the article.

Author Response

1) The article is mainly a catalog of recent MRI contrast agents. While this is of interest the manuscript would greatly improve if the author would critically comment and discuss drawbacks and advantages of different conjugates and chemistries in some more detail. A multitude of biological activities and potential applications are mentioned and the manuscript would greatly improve if a handfull representative examples would be described in more detail.

So far, all studies have stalled in the preclinical phase for reasons we described in the outlook. However, a lot of interesting science has been collected in the studies in question. In the current review, we aimed to catalogue that knowledge. In our opinion, it is undesirable to extend this review with a handful of detailed descriptions. The interested readers can find that information easily in the references provided. Moreover, such an extension would greatly increase the size of this manuscript, which is already quite long.

2) Over time numerous glyco-MRI agents were synthesized, but few clinical applications of diagnostics exist. A critical discussion of the therapeutic potential would add value to the article.

Actually, to the best of our knowledge, no clinical applications of diagnostics exists. The reasons are probably the limitations by economic, toxicologic, and environmental factors. The agents that may have therapeutic or theranostic potential are already indicated in the review. The obstacles mentioned above will hamper the realization. Probably, a fundamental different approach is needed for the design of personalized medicines based on glycobiology. For example, by composing a toolbox with modules for targeting, reporting, and drug delivery that can be assembled at will by click reactions. The knowledge described in the review would be a good starting point for that.

Reviewer 2 Report

The authors presented the paper "MRI Contrast Agents in Glycobiology" Thank you for so interesting review. However, I have some minor comments

1) I think it will be great to mention sub effects and toxicity studies of Gd-complexes. Recent works show their possible toxicity. In this way, some new non-metal contrast agents were developed. The same things about reactive oxygen species production by Fe oxide nanoparticles. Toxicity of the probes is one of the most important topics.

2) I highly recommend presenting some information about possible toxicity, about withdrawing of some Fe oxide and Gd-helates. Moreover, I don't agree with so optimistic Outlook section about the only Gd future in the area. Only the most stable Gd-complexes works excellent. Please, write some word about the possible future of Fe oxide nanoparticles and non-metall MRI approaches in the Outlook section.

3) Fig. 1 caption. You should mention that this relaxivities in water or water solution. Multihance, Primovist can be bound by albumin protein in plasma. The relaxivities will greatly enhance.

4) Fig.8, 9, 10 too small. It is difficult to see anything. Can you enlarge them of the whole page width?

Author Response

The authors presented the paper "MRI Contrast Agents in Glycobiology" Thank you for so interesting review.

We thank the referee for the positive evaluation of our manuscript.

However, I have some minor comments

1) I think it will be great to mention sub effects and toxicity studies of Gd-complexes. Recent works show their possible toxicity. In this way, some new non-metal contrast agents were developed. The same things about reactive oxygen species production by Fe oxide nanoparticles. Toxicity of the probes is one of the most important topics.

This review addresses in some detail toxicity studies of Gd-complexes used as MRI CAs in the clinic in the second and part of the third paragraph of the Outlook section (page 34), including references 187 to 194. Some more detail and new references were included. The reactive oxygen species production by Fe oxide nanoparticles and the related toxicity issues are discussed (see point 2)

2) I highly recommend presenting some information about possible toxicity, about withdrawing of some Fe oxides and Gd-chelates.

For the toxicity issue of Gd-chelates, see point 1) above. For toxicity withdrawing of some Fe oxides, a phrase was included in page 35:

“Slowed clearance may for example give rise to reactive oxygen species mediated toxicity [205-208]. Moreover, CAs based on almost all iron oxide NPs are withdrawn from the market, probably for commercial reasons.”

3) Moreover, I don't agree with so optimistic Outlook section about the only Gd future in the area. Only the most stable Gd-complexes works excellent.

We included some phrases and references on page 35 about the development of Mn- and Fe(III) based small chelates and NPs besides iron oxides, which summarise the huge amount of recent publications on these subjects. Toxicity issues of these are also referred to.

4) Please, write some words about the possible future of Fe oxide nanoparticles and non-metal MRI approaches in the Outlook section.

The future of Fe oxide nanoparticles depends on market issues, see point 2), in the phrase

“Moreover, CAs based on almost all iron oxide NPs are withdrawn from the market, probably for commercial reasons.”

The possible future of non-metal MRI approaches based on glucoCEST is limited by the low sensitivity of the technique. This is explained in reference 7.

5) Fig. 1 caption. You should mention that this relaxivities in water or water solution. Multihance, Primovist can be bound by albumin protein in plasma. The relaxivities will greatly enhance.

The relaxivities as measured in biological medium are now included between brackets in Fig. 1. The caption is adapted accordingly.

6) Fig.8, 9, 10 too small. It is difficult to see anything. Can you enlarge them of the whole page width?

This was done.